# Low-temperature pressure-assisted liquid-metal printing for $\beta$-Ga$_2$O$_3$ thin-film transistors

Chi-Hsin Huang[1], Ruei-Hong Cyu[2], Yu-Lun Chueh [2,3,4,5] & Kenji Nomura [1,6] ✉

Developing a low-temperature and cost-effective manufacturing process for energy-efficient and high-performance oxide-thin-film transistors (TFTs) is a crucial step toward advancing next-generation device applications such as wearable and flexible electronics. Among several methods, a liquid-metal printing technique is considered a promising, cost-effective oxide semiconductor process due to its inherent advantages, such as vacuum-free, low-thermal budget, high throughput, and scalability. In this study, we have developed a pressure-assisted liquid-metal printing technique enabling the low-temperature synthesis of polycrystalline wide bandgap n-channel oxide-TFTs. The n-channel oxide TFTs based on ~3 nm-thick $\beta$-Ga$_2$O$_3$ channels exhibited good TFT switching properties with a threshold voltage of ~3.8 V, a saturation mobility of ~11.7 cm$^2$ V$^{-1}$ s$^{-1}$, an on/off-current ratio of ~10$^9$, and a subthreshold slope of ~163 mV/decade. We also observed p-channel operation in the off-stoichiometric GaO$_x$ channels fabricated at high-pressure conditions. Toward oxide-based circuit applications, we developed high-performance oxide-TFT-based inverters. While our approach can promote the advancement of low-temperature manufacturing for oxide TFT technology, further work will be necessary to confirm the role of the applied pressure in the $\beta$-Ga$_2$O$_3$ crystallization process.

The demand for sustainable high-performance electronic devices and integrated circuits has been on the rise, especially for advanced device applications such as flexible displays[1], lightweight wearable electronics and sensors[2,3], electronic skin[4,5], Internet of Things[6], and back-end-of-line (BEOL)-compatible transistors for three-dimensional highly-integrated circuits[7]. To develop these device applications, it is crucial to explore low-temperature semiconductor device fabrication processes that not only reduce production costs but also minimize the environmental impact of electronic devices, significantly contributing to the development of sustainable electronics. Therefore, enormous efforts to develop semiconductor materials and device processing have been devoted to paving the way for the realization of environmentally friendly and cost-effective sustainable electronics[8–16].

In recent years, a variety of semiconductor materials such as organics and oxide semiconductors have been developed for low-temperature and cost-effective processing such as solution process and ink-jet printing[8–15,17]. Among these materials, ionic metal-oxide semiconductors, composed of post-transition metals, such as a-In-Ga-Zn-O (a-IGZO), In$_2$O$_3$, ZnO, SnO$_2$ have emerged as promising electronic materials to develop low-temperature processed and cost-effective,

[1]Department of Electrical and Computer Engineering, University of California San Diego, La Jolla, CA 92093, USA. [2]Department of Materials Science and Engineering, National Tsing Hua University, Hsinchu 30013, Taiwan. [3]College of Semiconductor Research, National Tsing-Hua University, Hsinchu 30013, Taiwan. [4]Department of Physics, National Sun Yat-Sen University, Kaohsiung 80424, Taiwan. [5] Department of Materials Science and Engineering, Korea University, Seoul 02841, Republic of Korea. [6]Material Science and Engineering Program, University of California San Diego, La Jolla, CA 92093, USA. ✉e-mail: kenomura@ucsd.edu

ubiquitous electronics. In general, it is well-known that structural defects are a common issue in low-temperature material/device processing, causing severe degradation of electrical properties in conventional semiconductors. However, n-type oxide semiconductors are less sensitive to structural defects due to their unique electronic structure nature, and electron transport is not significantly hindered[18]. This advantage offers a great opportunity to develop low-temperature processed electronic devices.

Currently, there have been significant advancements in the development of solution-processed oxide-based thin-film transistors (oxide TFTs), leading to several high-performance n-channel oxide TFTs[19-25]. However, an additional device fabrication process, such as post-thermal annealing, is generally required to achieve a device-quality semiconducting channel for TFT applications. A post-annealing process for oxides typically involves relatively high temperatures (usually between 300–600 °C) and can potentially negate the advantages of oxide semiconductors for low-temperature processed electronic devices[19-22,24,25]. Additionally, precise control of the processing atmosphere is often necessary, even in solution processes[25,26]. As a result, there is a high demand for oxide-based device processing at low temperatures, vacuum-free, and non-controlled atmospheric circumstances.

$\beta$-gallium oxide ($\beta$-Ga$_2$O$_3$) is a well-known n-type wide-bandgap oxide semiconductor with ultra-large wide-bandgap nature of 4.4–4.9 eV and relatively high electron mobility of 100–200 cm$^2$ V$^{-1}$ s$^{-1}$[27-30] for a wide field of device applications such as deep-ultraviolet (DUV) optoelectronics[31-33], power electronics operated in high-electrical fields (~MV/cm) and high-temperature devices (250–300 °C)[34-36]. The wide-bandgap nature of Ga$_2$O$_3$ is especially advantageous for designing devices capable of operating in harsh environments, such as extreme environment sensors, space-based applications, and terrestrial applications. Additionally, $\beta$-Ga$_2$O$_3$ is expected to develop energy-efficient operable transistors with low power consumption due to the low-channel leakage (i.e., off-current) nature originating from the low intrinsic carrier concentration and wide-bandgap. Therefore, Ga$_2$O$_3$ holds promise as a semiconductor material for microelectronics, and several demonstrations of high-performance $\beta$-Ga$_2$O$_3$ transistors have already been achieved. Current research on $\beta$-Ga$_2$O$_3$ transistors primarily focuses on metal-oxide-semiconductor field-effect transistor (MOSFET) structures using bulk single-crystal wafers for power devices. However, developing $\beta$-Ga$_2$O$_3$ TFTs remains challenging due to the high-temperature process required for $\beta$-Ga$_2$O$_3$ film growth in conventional physical vapor deposition (PVD) methods. Nevertheless, low-temperature processes for Ga$_2$O$_3$ TFTs show promise for various practical applications, such as low-power switching devices and solar-blind deep-ultraviolet photodetectors, originating from their ultra-wide-bandgap nature.

To date, high-performance Ga$_2$O$_3$ transistors use electron-doped single-crystal $\beta$-Ga$_2$O$_3$ channels, grown at a high temperature of >600 °C[34,35,37,38] and prepared using a mechanical exfoliation process[39-41]. Recently, amorphous and polycrystalline $\beta$-Ga$_2$O$_3$ TFTs have also been demonstrated, but achieving good TFT device operations still requires high process temperatures (200–900 °C)[42-45]. Moreover, the mobility of the previously reported Ga$_2$O$_3$ TFTs is only limited to <2 cm$^2$ V$^{-1}$ s$^{-1}$, which is insufficient for high-performance device applications[42-45]. Thus, developing a low-temperature process for $\beta$-Ga$_2$O$_3$ thin film with high mobility is critical to advancing the field of ultra-wide-bandgap oxide semiconductor electronic devices. Furthermore, Ga$_2$O$_3$ emerges as the next promising channel material in TFT technology to significantly reduce off-current and power consumption compared to the currently commercialized a-IGZO TFTs.

In this work, we proposed a solvent-free, vacuum-free, pressure-assisted liquid metal printing method called PA-LMP to fabricate high-performance $\beta$-Ga$_2$O$_3$ TFTs. The liquid metal printing (LMP) method is proposed for forming high-quality crystalline oxide materials with superior electrical properties compared to conventional physical/chemical vapor thin-film deposition, owing to its distinct thermodynamic pathway[17,46]. Therefore, the LMP method has already been demonstrated to develop high-mobility Indium-based n-channel oxide TFTs[46]. Here, the PA-LMP approach, which involves applying external uniaxial pressure during oxide skin formation on liquid metal, is demonstrated to directly form high-quality polycrystalline $\beta$-Ga$_2$O$_3$ nanosheets at a low temperature of 150 °C under a non-controlled ambient air atmosphere, exhibiting high-performance transistor operation. The presented n-channel $\beta$-Ga$_2$O$_3$ TFTs exhibited superior electrical characteristics with a high saturation mobility of 11.7 cm$^2$ V$^{-1}$ s$^{-1}$, on/off-current ratio of ~10$^9$, and subthreshold swing of 163 mV·decade$^{-1}$. Additionally, we successfully demonstrated a zero-$V_{GS}$-load NMOS inverter composed of enhancement/depletion-mode $\beta$-Ga$_2$O$_3$ TFTs and an all-oxide-based CMOS inverter using liquid metal printed p-channel SnO TFTs. These inverters operated in rail-to-rail full-swing voltage transfer operations with high voltage gain and low static power dissipation.

## Results and discussion

### Liquid-metal printing for Ga$_2$O$_3$ growth and characterization

A family of Ga$_2$O$_3$ is known to consist of five crystalline polymorphic forms, namely $\alpha$, $\beta$, $\gamma$, $\delta$, and $\varepsilon$ phases. The monoclinic $\beta$-Ga$_2$O$_3$ phase is the most stable, with lattice constants of $a = 1.22$ nm, $b = 0.30$ nm, and $c = 0.58$ nm, and it exhibits good thermal stability[33,47] (as shown in Fig. 1b). Figure 1b summarizes the transformation relationships among the five polymorphic forms of Ga$_2$O$_3$ under equilibrium conditions. The $\alpha$ and $\gamma$ forms are metastable and can be easily formed from GaO hydrate and gel forms and then transform into $\beta$-phase. However, to achieve the $\beta$-Ga$_2$O$_3$ phase from these metastable phases, a high-temperature process of at least 300 °C is required[48].

On the other hand, $\beta$-Ga$_2$O$_3$ thin film can be directly grown by several PVD techniques, such as sputtering and pulsed laser deposition. However, high-temperature substrate heating and post-deposition annealing, typically over 500 °C, are required for the thin-film growth process[44,49]. These results imply that low-temperature growth of $\beta$-Ga$_2$O$_3$ is very challenging in conventional thin-film growth approaches. The liquid metal printing approach can form crystalline and highly conductive oxide films upon deposition, avoiding insulating intermediate phases and eliminating the thermodynamic barriers posed by precursor decomposition[17,46], which is typical in the traditional approach to growing $\beta$-Ga$_2$O$_3$. The traditional processes involve the formation of GaO hydrate and gel forms, which then transform into the $\beta$-phase during subsequent processing. (Fig. 1b) We believe the oxidation process of the liquid metal presents different thermodynamic barriers and pathways for growing oxide materials, potentially making it easier to produce high-quality oxide materials at lower process temperatures. Furthermore, additional energy sources to modify thermodynamic parameters for crystallization are needed to overcome this issue and lower the growth temperature. It is considered that the external environmental pressure is a typical fundamental thermodynamic parameter and kinetic variable for the free energy change of the materials system defined by $\Delta G_f = \Delta(E + PV) - T\Delta S$, where $E$ is internal energy, $P$ is pressure, $V$ is volume, $T$ is temperature, and $S$ is entropy[50]. External pressure provides the opportunity to facilitate the crystallization process of oxide materials and lowers the crystallization temperature[51]. Therefore, we developed a method for low-temperature growth of $\beta$-Ga$_2$O$_3$ using external uniaxial pressure-liquid metal printing.

Figure 1c shows a schematic illustration of the PA-LMP method for fabricating Ga$_2$O$_3$ nanosheets under ambient air. The liquid-metal printing process was performed at 150 °C under an external uniaxial pressure of approximately 29 kPa. (printing process temperatures (T$_p$) of 150 °C and process pressure (P$_p$) of 29 kPa) The process steps are described in detail in Supplementary Fig. 1. The nanometer-thin surface oxide of liquid Ga is formed by self-limiting Cabrera−Mott

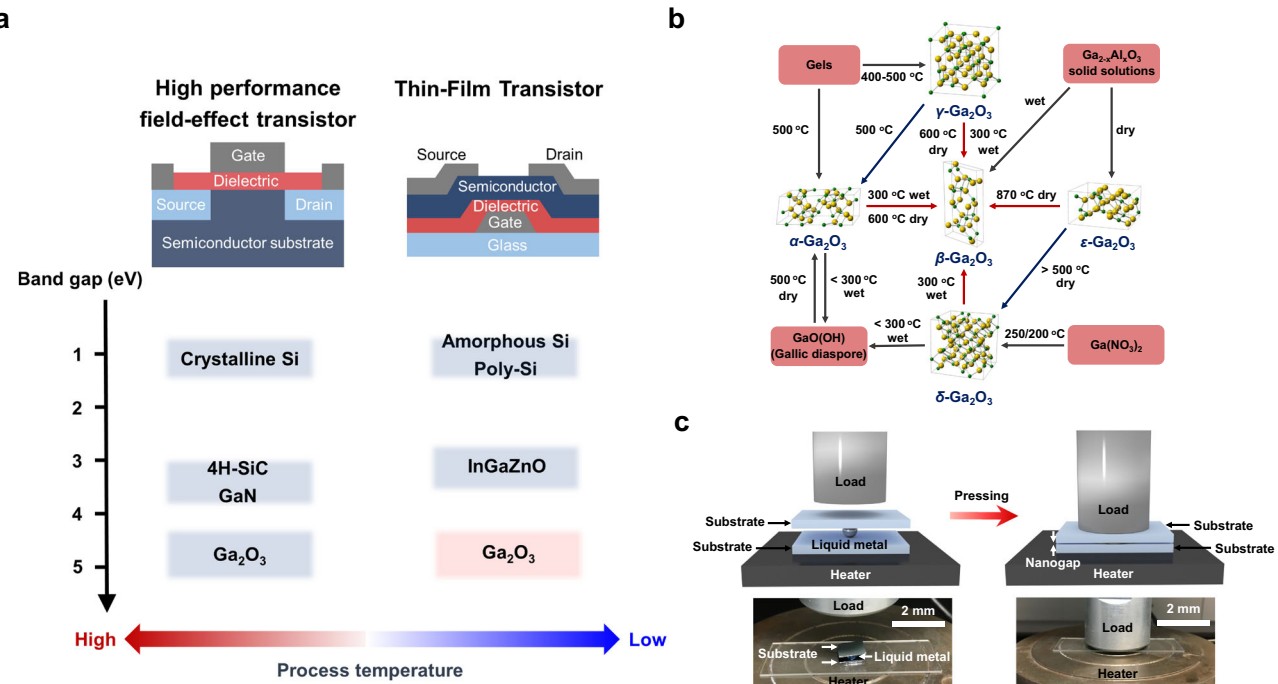

**Fig. 1 | Pressure-assisted liquid-metal printing for β-Ga₂O₃ thin-film transistors (TFTs). a** Motivation and purpose for developing low-temperature processed wide-bandgap Ga₂O₃ TFTs. **b** Phase transformation relationship for Ga₂O₃ polymorphism. **c** Schematic and photographs of the developed pressure-assisted liquid metal printing routes for Ga₂O₃ nanosheet fabrication.

oxidation in the ambient air to grow the Ga₂O₃ nanosheets. The optical microscope images and photographs of the Ga₂O₃ grown by previously reported LMP[52–55] and the proposed PA-LMP method show that these samples exhibit large-scale Ga₂O₃ nanosheets with dimensions of around 1 × 1 cm². (Fig. 1a, b) (The schematic of the conventional LMP is shown in Supplementary Fig. 2) The entire laterally large Ga₂O₃ layer is continuous and without significant holes and cracks under optimized synthesis operations. (Supplementary Fig. 3 shows additional optical microscope images of the Ga₂O₃ nanosheet fabricated by the different printing process parameters.) Fig. 2a, b displays an atomic force microscopy (AFM) image of nanosheets grown by the previously reported and proposed PA-LMP method. The edge step-height profile of these nanosheets found that the thickness was approximately 3.0 nm for both films. The root-mean-square of surface roughness ($R_{RMS}$) of the Ga₂O₃ nanosheets grown using LMP and PA-LMP methods were measured to be approximately 1.16 and 0.64 nm, respectively, comparable to the $R_{RMS}$ of SiO₂ surface (approximately 1.0 nm). These results suggest that the surface roughness of the Ga₂O₃ nanosheets is primarily due to the underlying SiO₂/Si substrate. Moreover, the presented Ga₂O₃ nanosheets were homogeneously and conformally grown on the substrates. Therefore, it was also confirmed that the developed pressure-assisted liquid-metal printing enabled the direct printing of large-area oxide materials onto various substrates, including SiO₂/Si, III-V (GaAs), transparent glass, paper, and flexible plastic substrates like PET, allowing the development of versatile applications by integrating diverse functional materials and substrates (Supplementary Fig. 4).

We fabricated inverted-staggered (i.e., bottom-gate and top-contact) structured TFT devices using the Ga₂O₃ nanosheets as a channel layer on thermally-oxidized SiO₂/p⁺-Si substrate (Fig. 2c). The SiO₂ served as the gate oxide with a thickness of 150 nm, while the p + -Si acted as the gate electrode. The ITO was used as the source/drain electrode for the Ga₂O₃ TFTs to form ohmic contacts[33,56,57]. The channel width ($W$) and length ($L$) are 300 and 100 μm, respectively. The optical microscopy image and scanning electron microscopy (SEM)

image of the Ga₂O₃ TFT are shown in Supplementary Fig. 5. Figure 2d displays the transfer characteristics of the Ga₂O₃ nanosheet TFTs fabricated by the conventional LMP and PA-LMP methods. In the case of the Ga₂O₃ fabricated by conventional LMP, the drain current was comparable to the background current level (-pA) of our device measurement system, and no TFT action was observed. This indicates that the LMP-grown nanosheet is electrically insulating.

In contrast, the PA-LMP-grown Ga₂O₃ channel ($T_p$ of 150 °C and $P_p$ of 129 kPa) exhibits desirable n-channel TFT operation, in which the drain currents ($|I_{DS}|$), measured by the drain-to-source voltage ($V_{DS}$), increase upon applying negative gate bias ($V_{GS}$). The output characteristics show ohmic liner-relations at small $V_{DS}$ regions and clear pinch-off behavior with the $I_{DS}$ saturation, confirming that the device operation follows the standard MOSFET model (Fig. 2e, f). The key TFT device characteristics (i.e., saturation mobility ($\mu_{sat}$), linear mobility ($\mu_{lin}$), s-value, and threshold voltage ($V_{th}$)) were determined as the following: the $\mu_{sat}$, $\mu_{lin}$ are estimated by the following equation, $|I_{DS}| = \mu_{sat} C_{ox} (\frac{W}{L})(V_{GS} - V_{th})^2$, $|I_{DS}| = \mu_{lin} C_{ox} (\frac{W}{L})[(V_{GS} - V_{th}) V_{DS} - \frac{1}{2} V_{DS}^2]$, respectively, where $C_{ox}$ is the gate insulator capacitance per unit area. The subthreshold slope (s-value) is extracted from the slope in the semi-logarithmic plot with $s = (\partial \log I_{DS} / \partial V_{GS})^{-1}$. A reasonable high $\mu_{sat}$ of 11.7 cm² V⁻¹ s⁻¹ and $\mu_{lin}$ of 10.2 cm² V⁻¹ s⁻¹ was obtained for the PA-LMP fabricated TFTs. The $V_{th}$ is determined from the intercept of a straight-line fit of the $(I_{DS})^{1/2} - V_{GS}$ plot and estimated 3.8 V, indicating that the device operates in enhancement mode, the "normally-off" operation, which is preferred for low-power analogs/digital circuits. The device also exhibits a small s-value of 163 mV dec⁻¹. Trap-state density ($D_{it}$) is estimated from the s-values using the following relation: $s = \frac{\log_e 10 \cdot k_B T}{e} \left[1 + \frac{e D_{it}}{C_{ox}}\right]$, where $e$ is the elementary electric charge, $k_B$ is the Boltzmann constant, $T$ is the temperature, $C_{ox}$ is the gate capacitance per area. The $D_{it}$ was estimated as $2.53 \times 10^{11}$ cm⁻² eV⁻¹, which is a significant improvement over previously reported Ga₂O₃ TFTs with the $D_{it}$ of -10¹² cm⁻² eV⁻¹. Importantly, the presented β-Ga₂O₃ nanosheet TFTs exhibit higher mobility and lower $D_{it}$ with a much lower process temperature compared to reported Ga₂O₃ TFTs fabricated by traditional thin-film processing. (Supplementary Table 1 for the summary

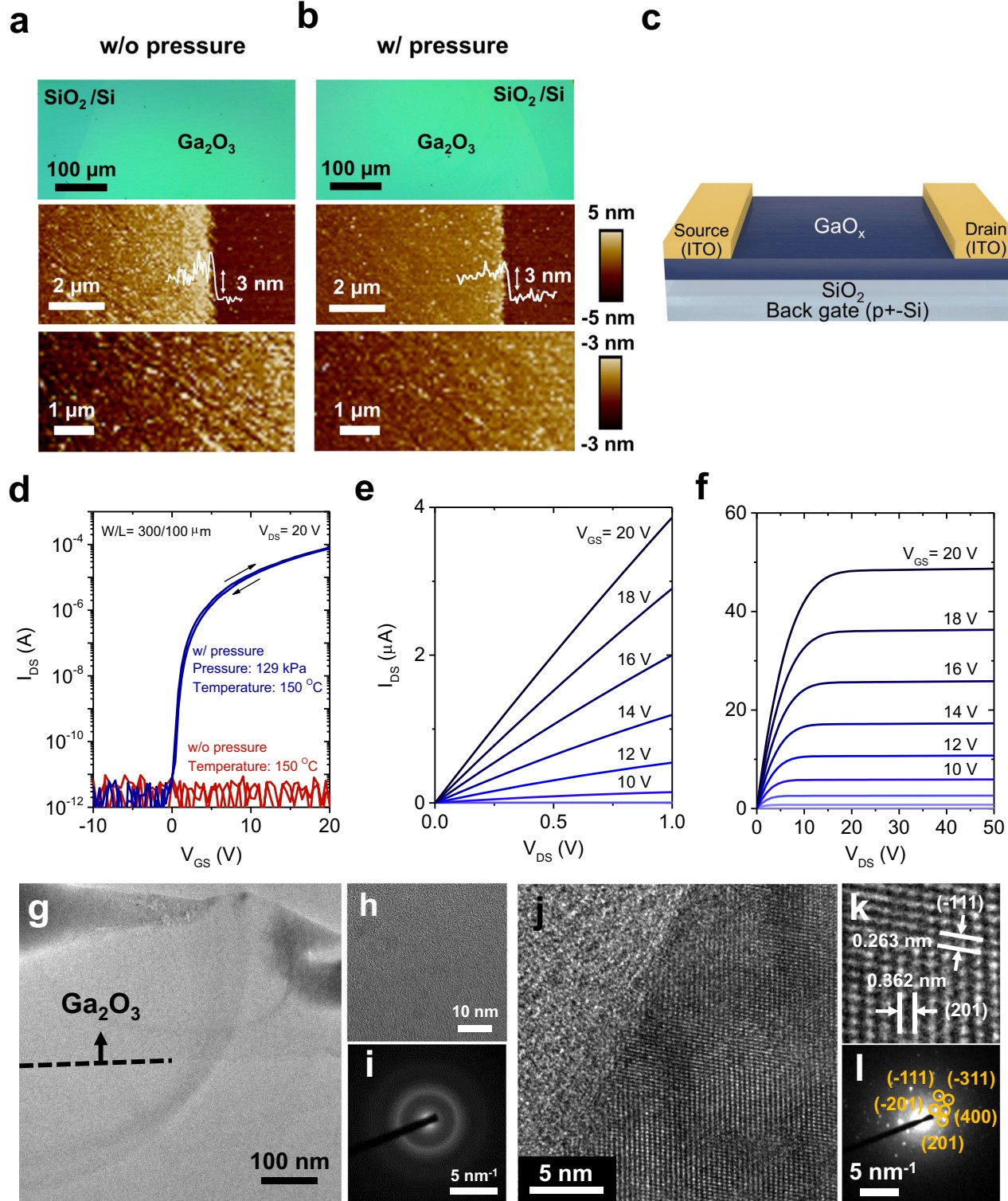

**Fig. 2 | Material and electrical characterization of Ga₂O₃ films.** Optical microscopy image and atomic force microscopy (AFM) image (including cross-sectional step-height profile) for the Ga₂O₃ nanosheet were prepared by **a** conventional liquid metal printing (LMP) (without applied printing process pressure) and **b** pressure-assisted liquid metal printing (PA-LMP) methods (applied printing process temperatures ($T_p$) of 150 °C and printing process pressure ($P_p$) of 129 kPa). **c** Schematics of the device structure of the GaO$_x$ thin-film transistors (TFTs). **d** Typical transfer characteristics for the Ga₂O₃ TFTs prepared by LMP ($T_p$ of 150 °C and $P_p$ of 0 kgf/cm²)) and PA-LMP ($T_p$ of 150 °C and $P_p$ of 129 kPa) methods. ($I_{DS}$: drain currents, $V_{DS}$: drain-to-source voltage, $V_{GS}$: gate bias) The channel width ($W$)

and length ($L$) of TFTs are 300 and 100 μm, respectively. **e** Magnified view of the linear regime in the typical output characteristics for the Ga₂O₃ TFT. **f** Typical output characteristics for the Ga₂O₃ TFT. **g** Low-magnified high-resolution transmission electron microscopy (HRTEM) images of the Ga₂O₃ nanosheet synthesized by the conventional LMP route. (without applied printing process pressure) Corresponding **h** high-magnified HRTEM image and **i** selected area electron diffraction (SAED) pattern. **j** Low-magnified HRTEM images of Ga₂O₃ nanosheet synthesized by the pressure-assisted liquid-metal route. ($T_p$ of 150 °C and $P_p$ of 129 kPa) Corresponding **k** high-magnified HRTEM image and **l** SAED pattern.

of device performance for previously reported $Ga_2O_3$ TFTs). These findings demonstrate that the PA-LMP method produces a high-quality $Ga_2O_3$ semiconductor nanosheet with a low defect density at a low temperature. In addition, the off-current density of the presented TFT devices is estimated to be $\sim 3 \times 10^{-15}$ A/μm, limited by the measurement instrument. The leakage current of the $Ga_2O_3$ TFTs is potentially even lower than that of commercial a-IGZO TFT devices ($\sim 10^{-18}$ A/μm)[58,59] in display applications. This is attributed to the wider bandgap nature of the $Ga_2O_3$ channel compared to a-IGZO. The device-to-device statistical analysis was also performed using ten working devices on a single substrate, and the results are summarized in Supplementary Fig. 6. The average values of $\mu_{sat}$ of $5.02 \pm 4.05$ cm$^2$ V$^{-1}$ s$^{-1}$, $\mu_{lin}$ of $4.21 \pm 5.87$ cm$^2$ V$^{-1}$ s$^{-1}$ s-values of $0.19 \pm 0.12$ V·dec$^{-1}$, $V_{th}$ of $3.61 \pm 2.99$ V, and log($I_{on}/I_{off}$) of $8.09 \pm 1.54$ were obtained. (Supplementary Figs. 6–8 provides a detailed discussion about the device uniformity) Furthermore, we observed no significant degradation in the device characteristics of the $Ga_2O_3$ TFTs even after 1 month of storage under a non-controlled ambient atmosphere, confirming the environmental stability of the devices in the air. (Supplementary Fig. 9 for the environmental stability test for $Ga_2O_3$ TFTs) The photoresponse property of $Ga_2O_3$ TFTs was also evaluated. No photoresponse was observed under green, blue, or UV light illuminations, which can be attributed to the wide-bandgap nature of $Ga_2O_3$. (Supplementary Fig. 10).

From the output characteristics, the output resistance of $1/g_d$, where $g_d$ is the output conductance, was evaluated as >80 MΩ (Supplementary Fig. 11a). We also calculated the transconductance, $g_m$, and the $1/g_d$ as a function of $V_{GS}$ (Supplementary Fig. 11b). The intrinsic gain, defined as $A_i = g_m/g_d$, was as high as 1000 at $V_{DS}$ of 30 V, significantly higher than that for other source-gate oxide TFTs (Schottky-barrier oxide TFTs)[60–62] and one order of magnitude higher than that for traditional ohmic contact IGZO TFTs. (Supplementary Fig. 11c) The high gains in the presented $Ga_2O_3$ TFTs guarantee the high potential of amplifier applications with better circuit stability and signal-to-noise ratio in digital/analog circuits. Moreover, good current saturation characteristics are immune to wide-range $V_{DS}$ modulation, making it a promising candidate for use as a current source in pixel circuits.

We also investigated the effect of post-thermal annealing atmospheres on the ultrathin $Ga_2O_3$ TFTs fabricated by PA-LMP. We observed that the TFT characteristics remain almost unchanged during annealing up to a temperature of 200 °C. (Supplementary Fig. 12). On the other hand, vacuum annealing ($\sim 10^{-5}$ mTorr) improved the TFT mobility but also exhibited a negative $V_{th}$ shift with the increase of off-current. This observation is attributed to the generation of extra carriers due to oxygen vacancy formation by vacuum annealing.

We analyzed the nanosheet structures grown by the conventional LMP and PA-LMP methods using transmission electron microscopy (TEM). (TEM sample preparation procedure was provided in Supplementary Fig. 13). Figure 2g shows a TEM image of the $Ga_2O_3$ nanosheet grown by the LMP method. The corresponding high-resolution TEM (HRTEM) image reveals no lattice-ordered structure in the LMP-grown nanosheet. (Fig. 2h) The corresponding selected area electron diffraction (SAED) pattern confirmed a halo pattern (Fig. 2i). These observations conclude that the nanosheet prepared by the LMP method is amorphous, which agrees with the previous reports[52,63–65]. Since amorphous semiconductor includes high-density tail-state defects[13], the nanosheet is speculated to involve high-density electron traps. Therefore, the LMP-grown amorphous GaO$_x$ exhibited a high-resistive state. We also attempted to improve the electrical properties of the amorphous GaO$_x$ nanosheets by post-thermal annealing at temperatures ranging from 200 to 400 °C. However, all devices remained insulative in the presented annealing conditions (Supplementary Fig. 14). To achieve a crystalline $\beta$-$Ga_2O_3$ phase, high-temperature annealing at temperatures above 500 °C may be required, as reported in previous studies[44,64,66]. However, high-temperature annealing may cause serious off-chemical

stoichiometry, which can lead to the failure of TFT operations in the presented nanosheet channel.

In contrast, a distinct polycrystalline feature, i.e., ordered atomic alignment, for the PA-LMP-grown nanosheet ($T_p$ of 150 °C and $P_p$ of 129 kPa) was observed in the TEM analysis (Fig. 2j and Supplementary Fig. 15a for the low-magnified HRTEM images of the $Ga_2O_3$ nanosheet). The HRTEM image also showed the crystal lattice structure with internal spacings of $\sim 0.362$ and $\sim 0.263$ nm, which are assigned to the (201) and (−111) planes, respectively, of the monoclinic $\beta$-$Ga_2O_3$ crystal structure. (Fig. 2k) The corresponding SAED pattern is shown in Fig. 2i, and exhibits spots indexed to the (−111), (201), (−311), and (400) crystal plans. The observation concluded that the presented nanosheet was randomly oriented $\beta$-$Ga_2O_3$ crystals. In addition, we performed TEM analysis for the $Ga_2O_3$ nanosheet prepared using the PA-LMP approach with different process parameters to confirm the direct growth of $\beta$-$Ga_2O_3$ crystals. (Supplementary Figs. 15–17) The energy-dispersive X-ray spectroscopy (EDX) chemical composition mapping analysis also finds that the Ga/O atomic ratio is $\sim 0.64$, which is close to the ideal stoichiometry of $Ga_2O_3$ (Supplementary Fig. 18). The grazing incidence X-ray diffraction (GIXRD) analysis also supported that the $Ga_2O_3$ nanosheets prepared by the PA-LMP method were polycrystalline $\beta$-$Ga_2O_3$. (Supplementary Fig. 19) Furthermore, we conducted a cross-sectional TEM analysis for the $Ga_2O_3$ nanosheet prepared using the liquid metal printing approach on the Si/SiO$_2$. The $Ga_2O_3$ nanosheet was prepared at the $T_p$ of 150 °C, and the $P_p$ of 129 kPa, exhibiting a high TFT mobility of 8–10 cm$^2$ V$^{-1}$ s$^{-1}$. The distinct polycrystalline nature of $\beta$-$Ga_2O_3$ was also observed, providing direct evidence that liquid metal printing can grow crystalline $\beta$-$Ga_2O_3$ on the SiO$_2$/Si substrate. (Supplementary Figs. 20–23) X-ray photoemission spectroscopy (XPS) analysis also confirmed that this material is $Ga_2O_3$ with the Ga$^{3+}$ oxidation state. (Supplementary Fig. 24) Based on the material characterization discussed above, the developed PA-LMP method directly grows crystalline $\beta$-$Ga_2O_3$ with device quality at less than 150 °C.

We believe that the nature of oxidation of the liquid metal has a different thermodynamic barrier and path to grow oxide materials, potentially making it easier to grow high-quality oxide materials at lower process temperatures compared to other material growth approaches[17,46]. Crystalline $\beta$-$Ga_2O_3$ has been successfully synthesized using liquid metal printing, achieved by scraping off the parent metal layer under stress at a process temperature of 200 °C[67]. Given that the thermodynamic barrier and pathway may lower when employing liquid metal oxidation, the crystallization of the $Ga_2O_3$ phase could potentially be induced by various factors, such as heat, pressure, and stress, to lower the energy barriers for crystallization. Furthermore, our findings suggest that the effects of nanoscale confinement crystal growth and external pressure may influence the facilitation of low-temperature crystallization for the $\beta$-$Ga_2O_3$ phase. It is well-known that nucleation is the initial critical process for crystallization, so reducing the free energy barriers of the nucleation process is essential. In general, free energy barriers for homogeneous ($\triangle G_{homo}$) and heterogeneous nucleations ($\triangle G_{hetero}$) are defined by

$$\triangle G_{homo} = \frac{16\pi \gamma^3}{3(\triangle G)^2}, \triangle G_{hetero} = \triangle G_{homo} \cdot f(\theta) \tag{1}$$

Where $\gamma$ is the interfacial energy. $f(\theta)$ is a function related to the contact angle $\theta$ as

$$f(\theta) = \frac{2 - 3\cos\theta + \cos^3\theta}{4} \tag{2}$$

In the developed PA-LMP method, the contact angle is nearly zero since the liquid metal is confined in the nanogap that two substrates generate. Confined systems are expected to influence crystallization processes significantly. As the volume of the confining space

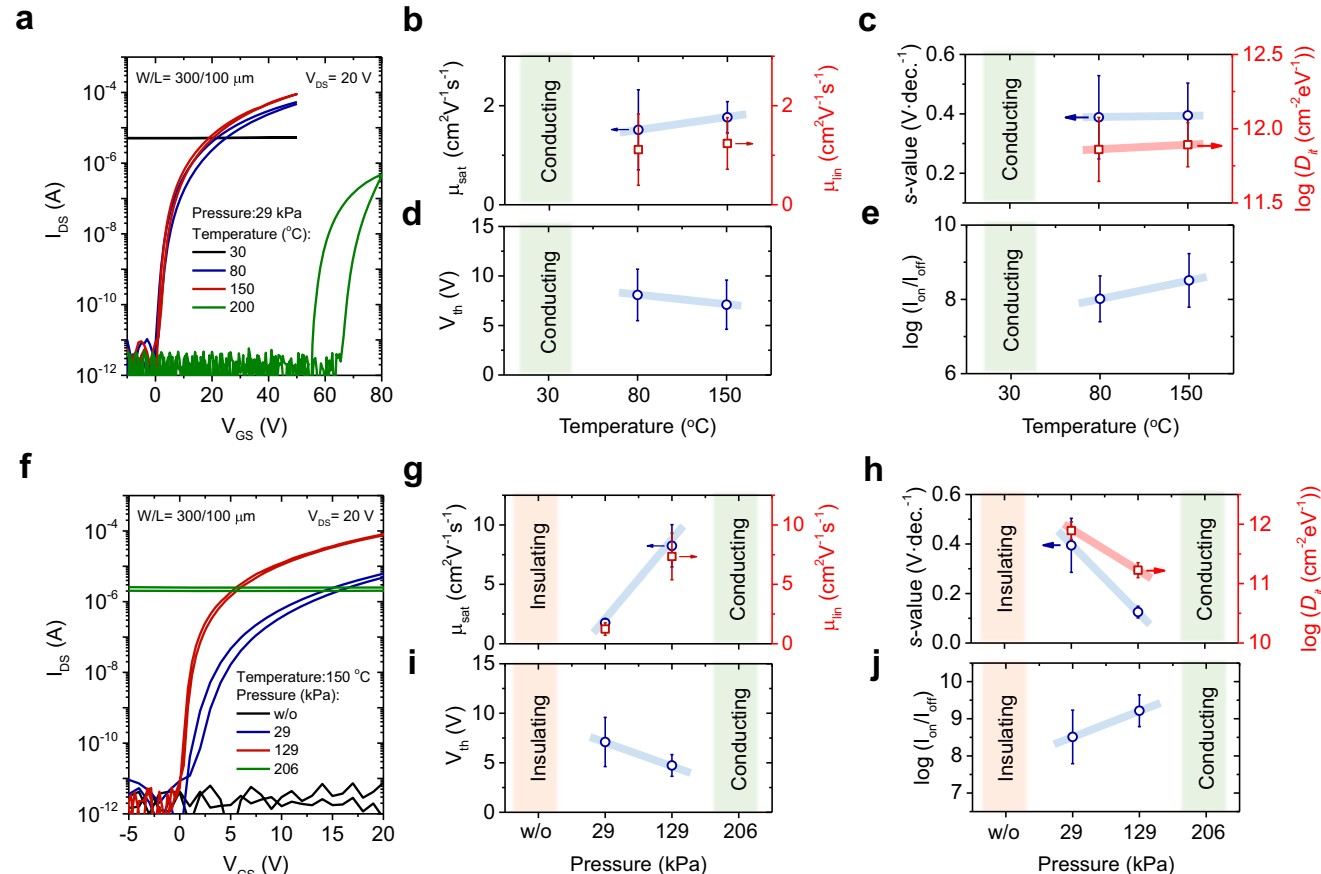

**Fig. 3 | Effect of the printing process on Ga₂O₃ thin-film transistor (TFT) performance. a** Variation of transfer characteristics for the Ga₂O₃ nanosheet TFTs with different printing process temperatures under the uniaxial process pressure of 29 kPa. The corresponding TFT parameters **b** saturation mobility ($\mu_{sat}$) and linear mobility ($\mu_{lin}$), **c** subthreshold swing (s-value) and trap-state density ($D_{it}$), **d** threshold voltage ($V_{th}$), and **e** on/off current ratio are plotted. **f** Variation of transfer characteristics for the Ga₂O₃ TFTs with different process pressures under the printing process temperature of 150 °C. Corresponding TFT parameters **g** $\mu_{sat}$ and $\mu_{lin}$, **h** s-value and $D_{it}$, **i** $V_{th}$, and **j** on/off current ratio are plotted. (The error bars are calculated using data from 16 representative working devices across different samples).

decreases, the role of the surface becomes increasingly essential in such constrained systems. To achieve low-temperature growth, crystallization in highly confined nanogap regions (nanoscale confinement) provides the potential approach to lower the energy barrier by reducing the reactive surface area of nuclei, thereby decreasing the surface energy penalty. Additionally, nanoscale confinement can alter the nucleation pathway, further reducing the overall energy barrier[68,69]. Previous studies have explored crystallization and phase transitions under nanoscale confinement[70–72]. For example, ref. 70 applied classical nucleation theory to investigate nucleation during the solidification and melting phase transitions of germanium (Ge) within nanoscale confinement between two planar surfaces. They found that nanoscale confinement significantly lowers nucleation temperatures of Ge for solid and liquid phases, with a 5.0 nm gap reducing nucleation temperature by up to 350 °C. A detailed discussion of two-dimensional crystallization in the nanoscale confined space is provided in the Supplementary Information.

Moreover, it is widely accepted that external pressure impacts the thermodynamic driving forces for crystal nucleation and growth processes. The pressure ($P$) on the nucleation activation energy, $\Delta G^*$, can be expressed as[73]

$$\left[\frac{\partial\left(\triangle G^*\right)}{\partial P}\right] = -\frac{32\pi\gamma^3}{3}\cdot\frac{\triangle V}{(\triangle G)^3} \qquad (3)$$

where $\triangle G = G_c - G_a$, where $G_c$ and $G_a$ are the Gibbs free energies of the crystalline and amorphous phases, respectively, $\gamma$ is interfacial energy and is not sensitive to pressure, $\triangle V = V_c - V_a$ is the difference in molar volumes between the crystalline phase ($V_c$) and the amorphous phase ($V_a$). Based on previously reported experimental results, the film density of amorphous GaOₓ was determined to be approximately 5.2–5.4 g·cm⁻³[43,74], which is lower than that of $\beta$-Ga₂O₃ (density of ~5.95 g·cm⁻³)[75–77]. In this case, i.e., $\frac{\partial(\triangle G^*)}{\partial P}<0$, since the smaller molar volume of the crystalline phase compared to the amorphous phase ($\triangle V = V_c - V_a <0$) and $\triangle G = G_c - G_a <0$. As a result, our findings suggest that pressure may also facilitate crystallization by potentially reducing the nucleation activation energy. Further research is required to validate the impact of applied pressure on the $\beta$-Ga₂O₃ crystallization process.

### Effect of the printing process on Ga₂O₃ transistor performance

Figure 3a illustrates the variation of transfer curves for the ultrathin $\beta$-Ga₂O₃ TFTs grown by the PA-LMP under a pressure of 29 kPa at different printing process temperatures. (Supplementary Fig. 26 shows the transfer characteristics for linear regions. ($V_{DS}=1$ V)) When the device was fabricated at 30 °C, the channel showed highly conducting behavior with an electrical conductivity of 28.3 S·m⁻¹, but the device exhibited negligible field-effect current modulation. Decent TFT actions with an on/off current ratio of 10⁵ were observed when the channels were grown at process temperatures above 80 °C. Furthermore, the TFT mobility was improved by raising the process

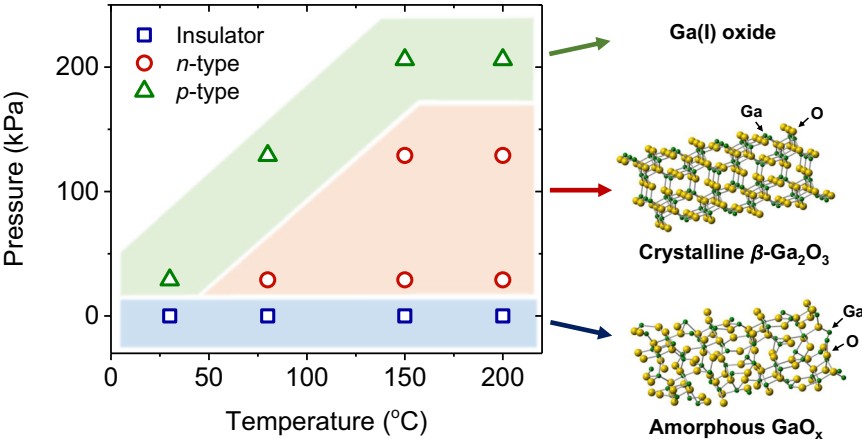

**Fig. 4 | Electrical properties of Ga$_2$O$_3$ films grown by liquid-metal printing.** Summary of the electrical properties of the gallium oxide nanosheet channel grown using the pressure-assisted liquid-metal route under various printing process conditions of uniaxial pressures and temperatures.

temperature up to 150 °C. However, a large positive $V_{th}$ shift to 58 V was observed for the device fabricated at a high temperature of 200 °C, indicating that the high-temperature process under a pressure of 29 kPa resulted in poor TFT performances. The fundamental device parameters ($\mu_{sat}$, $\mu_{lin}$, $s$-value, $V_{th}$, and on/off current ratio) are also summarized in Fig. 3b–e. The data were analyzed using 16 representative working devices from different samples.

The temperature at which the gallium oxide skin is formed on the Ga liquid surface metal plays a critical role in the oxidation process. Our observations suggest that the low process temperature of 30 °C may not be sufficient to form Ga$_2$O$_3$ phases using the PA-LMP method with a printing process pressure of 29 kPa, resulting in high-density Ga metal impurity and highly conductive channels in the as-prepared device. From the TEM characterization, it was also confirmed that Ga-rich GaO$_x$ with embedded Ga metal in nanosheets grown by the printing process at the T$_p$ of 30 °C and the P$_p$ of 29 kPa. (Supplementary Figs. 27, 28) Interestingly, the devices after post-thermal annealing showed p-channel operation with a small on/off current ratio (Supplementary Fig. 29). The origin of the p-type behavior is not yet clear, but we speculate that the p-type behavior may relate to low-valence Ga(I)-based metastable phases formation, such as GaO or Ga$_2$O, which possibly have the VBM structure composed of spherical Ga 5$s$ orbital. Further investigation of the origin of p-type behavior in gallium oxide is required.

On the other hand, a high-temperature printing process of PA-LMP is sufficient to oxidize gallium metal and form the Ga$_2$O$_3$ phase. However, the poor TFT performance was observed during the high-temperature process at 200 °C under a pressure of 29 kPa (T$_p$ of 200 °C and P$_p$ of 29 kPa). This is primarily caused by (1) the carrier concentration, resulting from oxygen vacancies, can be decreased through compensation by oxygen-containing annealing effect in the ambient air atmosphere, while (2) excessive oxygen defects, which function as acceptor-like defects, are introduced during oxygen-containing air annealing. Therefore, the optimal process temperature for $\beta$-Ga$_2$O$_3$ TFTs was found to be between 80–150 °C for the PA-LMP. We also fabricated TFTs using the channels prepared by the conventional LMP method at high temperatures of up to 200 °C and measured their electrical characteristics. However, no field-effect modulation was observed in any of the devices, indicating that the Ga$_2$O$_3$ fabricated LMP is electrically insulating regardless of the process temperature. (Supplementary Fig. 30).

Figure 3f shows the variation in transfer characteristics with different process pressures, with a process temperature of 150 °C. (Supplementary Fig. 26 also shows the transfer characteristics for linear ($V_{DS}$ = 1 V) region.) The performance of TFTs was found to improve with increasing external pressure, with the most significant improvement observed at uniaxial pressure of 129 kPa. The most optimized TFT switching properties were achieved under this condition, with high $\mu_{sat}$ of 11.7 cm$^2$ V$^{-1}$ s$^{-1}$, large on/off current ratio of ~10$^9$, and small subthreshold slope of 137 mV·decade$^{-1}$, and $V_{th}$ of ~4 V. The device exhibits clockwise hysteresis originating from electron traps, but the hysteresis window was improved to 0.1 V under high-pressure conditions (Supplementary Fig. 31). The key device parameters ($\mu_{sat}$, $\mu_{lin}$, $s$-value, $V_{th}$, and on/off current ratio) are also summarized in Fig. 3g–j. The data were analyzed using 16 representative working devices from different samples. The average values of $\mu_{sat}$ of 8.24 ± 1.78 cm$^2$ V$^{-1}$ s$^{-1}$, $\mu_{lin}$ of 7.34 ± 1.96 cm$^2$ V$^{-1}$ s$^{-1}$, $s$-values of 125 ± 23.6 mV dec$^{-1}$, $V_{th}$ of 4.73 ± 1.10 V, and log($I_{on}/I_{off}$) of 9.22 ± 0.43 were obtained under the temperature of 150 °C and the pressure of 129 kPa. (T$_p$ of 150 °C and P$_p$ of 129 kPa) At high-pressure conditions (206 kPa), the device operation switched from n-channel to p-channel devices, suggesting the formation of low valance Ga(I) under high-pressure conditions. The optical microscope images of these samples with different process conditions, which we investigated for their electrical properties, are also shown in Supplementary Fig. 3. Furthermore, the AFM image reveals the effect of the applied pressure on the surface morphology, as discussed in the Supplementary Figs. 32, 33.

Figure 4 summarizes the TFT operations fabricated by different process pressures and temperature conditions in the PA-LMP route. All The transfer characteristics are presented in Supplementary Fig. 34. Regardless of the nanosheet growth conditions, conventional LMP-grown nanosheets were electrically insulating amorphous gallium oxide, which was in good agreement with previous reports[64,78]. In contrast, the PA-LMP directly synthesized a semiconducting polycrystalline $\beta$-Ga$_2$O$_3$ channel in the range of the T$_p$ of 80–200 °C with the P$_p$ of 29 kPa and in the range of the T$_p$ 150–200 °C with the P$_p$ of 129 kPa. The polycrystalline nature of $\beta$-Ga$_2$O$_3$ is confirmed by both GIXRD and TEM characterization. (Supplementary Figs. 15–17, 19). The performance of TFT degraded with a positive threshold voltage shift during the high-temperature process at 200 °C under pressures of 29 and 129 kPa, respectively. The reason is suspected to be a decrease in carrier concentration originating from oxygen vacancies and the introduction of excessive oxygen defects functioning as acceptor-like defects during annealing in oxygen-containing air. Interestingly, amorphous films grown at high pressure of 206 kPa with the process temperature of 150 and 200 °C exhibited p-type behavior. We suspect these materials result from the slight oxidation of Ga metal, leading to off-stoichiometric GaO$_x$ materials. These materials might include low-valence Ga(I)-based metastable phases, possibly with a VBM structure composed of spherical Ga 5$s$ orbitals contributing to the *p*-type

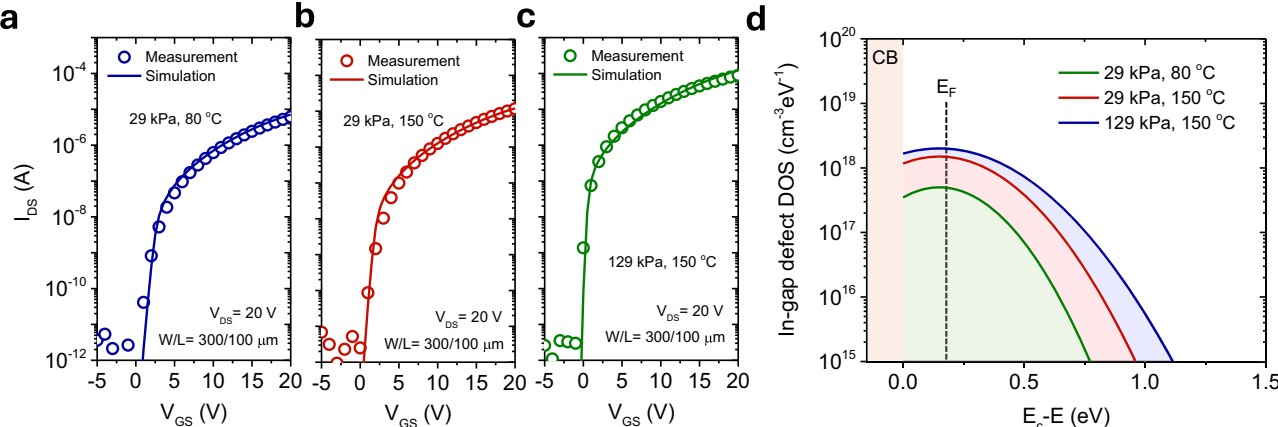

**Fig. 5 | TCAD simulation of Ga₂O₃ thin-film transistors (TFTs).** Comparison of measured (symbols) and simulated (lines) transfer curves with $V_{DS} = 20$ V for the Ga₂O₃ TFTs with different printing process conditions. (**a** $T_p$ of 80 °C, $P_p$ of 29 kPa, **b** $T_p$ of 150 °C, $P_p$ of 29 kPa, **c** $T_p$ of 150 °C, $P_p$ of 129 kPa). **d** Corresponding in-gap density of states (DOS) profiles for the Ga₂O₃ nanosheet channels for different printing process conditions. Fermi level ($E_F$) lies at $E_c - E_F = 0.23$ eV. ($E_c$: conduction band edge energy, CB:conduction band.) TCAD simulations were conducted to

extract the in-gap defect DOS profile in Ga₂O₃ TFTs. The measured I-V curves were accurately modeled by optimizing only the acceptor-like defects (electron trap states) using a Gaussian distribution, $g_G(E) = N_{GD} \cdot \exp\left\{ -\left[ \frac{(E_c - E_{GA})}{W_{GD}} \right]^2 \right\}$, where $N_{GD}$ is the state densities at the central energy $E_{GA}$ of the Gaussian distribution, $Ec$ is the conduction band edge energy for the reference zero point, $E_{GA}$ is the central energy of $g_G(E)$, $W_{GD}$ is the characteristic decay energy.

behavior. Further study is required to investigate these materials. We notice that there are reports of Ga ultrathin film growth using a similar liquid metal printing method[79,80]. These results corroborate our findings that liquid metal printing can form off-stoichiometric GaOₓ through slight oxidation of Ga metal under an air atmosphere.

To gain further insight into the variation of TFT characteristics, we performed the Technology Computer-Aided Design (TCAD) device simulation to extract the subgap defect density of the state (DOS) profile. Figure 5a–c shows the measured and simulated transfer curves of Ga₂O₃ TFTs with different process conditions. Only parameter optimization for the subgap acceptor-like defect density of state (DOS) can reproduce the measured transfer curves, indicating that the variation in TFT characteristics under different process parameters mainly originates from the change in subgap defect DOS in the Ga₂O₃ channels. The extracted subgap defect DOS near the conduction band (CB) of Ga₂O₃ TFTs under different process conditions are shown in Fig. 5d. (Supplementary Table 3 provides the parameters of the TFT simulations). We found that all the measured I-V curves were reproduced by only optimizing the acceptor-like defect (i.e., electron trap defect) with Gaussian distribution type,

$$g_G(E) = N_{GD} \cdot \exp\left\{ -\left[ \frac{(E_c - E_{GA})}{W_{GD}} \right]^2 \right\} \qquad (4)$$

where $N_{GA}$ is the state densities at the central energy $E_{GA}$ of the Gaussian distribution, $Ec$ is the conduction band edge energy for the reference zero point, $E_{GA}$ is the central energy of $g_G(E)$, $W_{GD}$ is the characteristic decay energy. Due to the polycrystalline nature of the Ga₂O₃, we opted not to employ the acceptor-like exponential DOS to fit the measured I-V curves that are used for the tail states near the conduction for the amorphous silicon (a-Si:H) and amorphous IGZO[81]. All the devices show a low carrier concentration of $5.5 \times 10^{14}$ cm³ due to the ultra-wide gap nature of Ga₂O₃.

We observed shallow acceptor-like defect states located at 0.15 eV below the conduction band ($Ec - 0.15$ eV) in the Ga₂O₃. TFTs. These acceptor-like defect states primarily function as electron traps in the n-channel oxide TFT devices. The physical origin of these shallow acceptor-like defect states remains elusive; however, we suspect these defects result from Ga vacancies or weakly bonded (excess) oxygen defects[43,82,83]. The Ga₂O₃ TFT fabricated with the low-pressure condition ($T_p$ of 80 °C, $P_p$ of 29 kPa) exhibited a relatively high density of

shallow acceptor-like defect states of $2 \times 10^{18}$ cm⁻³ eV⁻¹ and remained unchanged even in the device fabricated at 150 °C. (150 °C, 29 kPa) On the other hand, we found that the high-pressure condition (150 °C, 129 kPa) effectively reduced the shallow subgap acceptor-like defect DOS to $5 \times 10^{17}$ cm⁻³ eV⁻¹, resulting in better TFT performances with higher mobility. This acceptor-like defect reduction makes moving the Fermi level toward the mobility edge easier, achieving band-like conduction and explaining the higher mobilities observed in these TFTs.

We also found that the shallow subgap defect DOS significantly impacts the on-current and s-value but does not affect the turn-on voltage, resulting from the enhancement mode operation. The $D_{it}$, estimated from the s-values of the experimental transfer characteristics, are $5.37 \times 10^{11}$ cm⁻² eV⁻¹ (80 °C, 29 kPa), $4.78 \times 10^{11}$ cm⁻² eV⁻¹ (150 °C, 29 kPa), and $1.93 \times 10^{11}$ cm⁻² eV⁻¹ (150 °C, 129 kPa) for these three process conditions, respectively. The shallow subgap defect DOS for all these three conditions we attained from the TCAD simulation correspond to area densities of $6 \times 10^{11}$ cm⁻² eV⁻¹ (80 °C, 29 kPa), $4.5 \times 10^{11}$ cm⁻² eV⁻¹ (150 °C, 29 kPa) and $1.8 \times 10^{11}$ cm⁻² eV⁻¹ (150 °C,129 kPa), respectively, considering the Ga₂O₃ thickness of 3 nm. These values are consistent with those calculated from experimental s-values.

## Circuits based on Ga₂O₃ thin-film transistors

Since ultrathin oxide channels improve gate controllability and low-off current characteristics of the TFTs, developing a low-power inverter is highly expected for next-generation energy-efficient oxide electronics. We developed oxide-TFT-based inverter circuits, including NMOS and CMOS circuits, using ultrathin Ga₂O₃ TFTs. The Ga₂O₃ nanosheets used for the NMOS and CMOS circuits demonstration were grown using the $T_p$ of 150 °C and $P_p$ of 29 kPa. Figure 6a shows the typical transfer characteristics for a zero-$V_{GS}$ NMOS, consisting of the enhancement and depletion-mode TFTs as the driver and load, respectively. The depletion-mode TFTs were fabricated by performing post-thermal annealing under vacuum conditions at 100°C. The effect of vacuum annealing is discussed in Supplementary Fig. 12. The depletion-mode TFTs showed the $\mu_{sat}$ of 0.8 cm² V⁻¹ s⁻¹, s-value of 1.6 V·decade⁻¹, $V_{th}$ of −4.5 V, and on/off current ratio of ~10⁷. The corresponding TFT performances for enhancement-mode TFTs show the $\mu_{sat}$ of 2.6 cm² V⁻¹ s⁻¹, s-value of 0.28 mV·decade⁻¹, $V_{th}$ of 6 V, and on/off current ratio of ~10⁸, respectively. Figure 6b shows a typical voltage transfer characteristic (VTC) of the NMOS inverter, which is the $V_{out}$ as a function of the $V_{in}$ with $V_{DD}$. The inset illustrates the equivalent circuit diagram of the

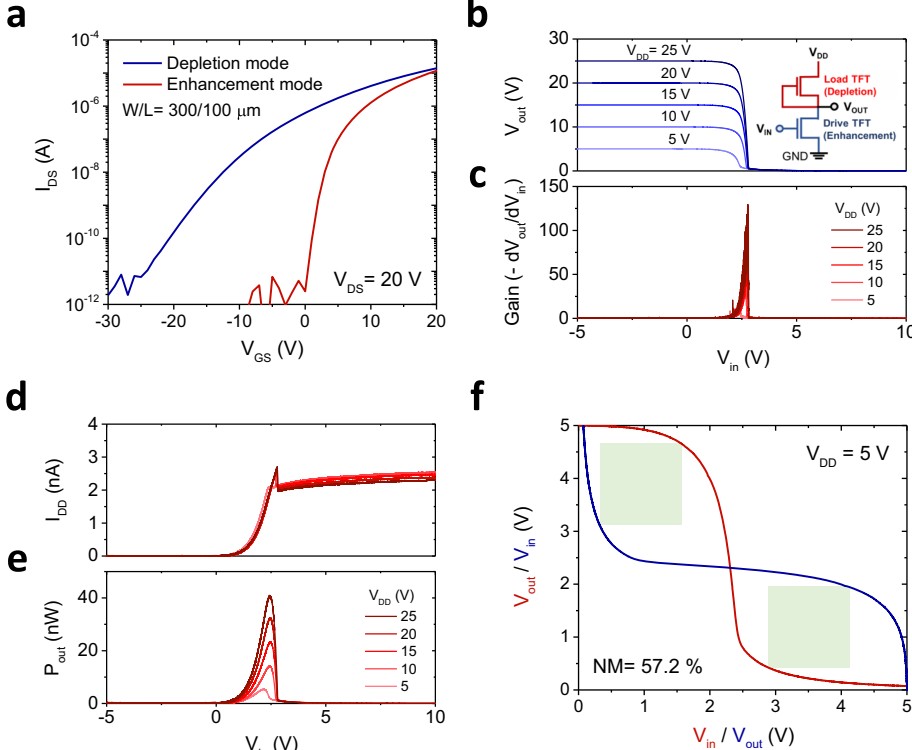

**Fig. 6 | Ga$_2$O$_3$ transistor-based N-channel Metal-Oxide-Semiconductor (NMOS) inverter.** An NMOS inverter is a logic circuit built using an n-channel MOSFET (NMOS transistor) and a pull-up resistor or load transistor. **a** Typical transfer characteristics for the depletion-mode and enhancement-mode Ga$_2$O$_3$ TFTs. (The Ga$_2$O$_3$ nanosheets used for the NMOS circuits demonstration were grown using the T$_p$ of 150 °C, P$_p$ of 29 kPa. The depletion-mode TFTs were fabricated by performing post-thermal annealing under vacuum conditions at 100 °C.) **b** Voltage transfer characteristic (VTC) of the Ga$_2$O$_3$ TFT-based NMOS inverter. **c** corresponding voltage gains of the Ga$_2$O$_3$ TFT-based NMOS inverter. Inset: schematic circuit diagram of the zero-$V_{GS}$-load NMOS inverter composed of enhancement/depletion-mode Ga$_2$O$_3$ TFTs. **d** Corresponding supply currents ($I_{DD}$). **e** Output-power consumption ($P_{out}$) for the Ga$_2$O$_3$ TFT-based NMOS inverter. **f** Butterfly curves for noise margin (NM) of the Ga$_2$O$_3$ TFT-based NMOS inverter. ($V_{in}$: input voltage, $V_{out}$: output voltage, $V_{DD}$: supply voltage).

zero-$V_{GS}$-load NMOS inverter. The output voltage switches from high to low, confirming a clear inverting action with the full-voltage swing. The average voltage gain of the NMOS inverters was 16.9, 45.1, 64.0, 80.5, and 106.4 at $V_{DD}$ from 5 to 25 V (Fig. 6c), respectively, comparable to the previously reported oxide-NMOS inverters[84–86] (Supplementary Fig. 35 for statistical results on the gain of the NMOS inverter, and Supplementary Table 4 for the NMOS inverter summary) Fig. 6d also shows the corresponding supply currents ($I_{DD}$) as functions of $V_{in}$. The static currents ($V_{in}$ = 0 V or $V_{in}$ = $V_{DD}$) are lower than 2.4 nA, and the static power dissipation, which is defined by $P_{static}$ = $V_{DD}$ ($I_{static\_low}$+$I_{static\_high}$)/2, is 28 nW per logic gate at $V_{DD}$ of 25 V. The $P_{out}$ is estimated as <43 nW per logic gate, confirming that a nanowatt power source can operate the presented inverter (Fig. 6e). The noise margin (NM) is estimated using the maximum equal criterion method and obtained at 57.2% of the ideal value ($V_{DD}$/2). This indicates that the device has sufficient NM for most static logic applications (Fig. 6f). The dynamic switching was also demonstrated to indicate the potential for circuit applications by further optimizing the dimensions of the devices. (Supplementary Fig. 36). The enhancement-load inverter, which consists of two enhancement-mode TFTs, was also demonstrated (Supplementary Fig. 37 for the detailed inverter performance).

All-oxide-CMOS inverter circuits were also developed using n-type ultrathin Ga$_2$O$_3$ TFT and $p$-type ultrathin SnO TFT, which is also fabricated by a liquid-metal printing technique. The detailed fabrication procedure of $p$-channel SnO TFTs can be found in our previous work[55]. The typical transfer characteristics of both the ultrathin $p$-channel SnO TFT and n-channel Ga$_2$O$_3$ TFT used in the oxide-TFT-based CMOS inverter are shown in Fig. 7a. The n-channel Ga$_2$O$_3$ TFT exhibits the $\mu_{sat}$ of 1.5 cm$^2$ V$^{-1}$ s$^{-1}$, $s$-value of 0.27 V dec.$^{-1}$, $V_{th}$ of 6 V, and on/off current

ratio of ~10$^8$. The corresponding TFT performances for $p$-channel SnO TFT are the $\mu_{sat}$ of 0.2 cm$^2$ V$^{-1}$ s$^{-1}$, $s$-value of 1.9 V·decade$^{-1}$, $V_{th}$ of −20 V, and the on/off current ratio of ~10$^5$, respectively. The inset of Fig. 7b is the schematic circuit diagram of the CMOS inverters. Figure 7b presents the typical voltage transfer characteristic of the CMOS inverter, where the output voltage switches from high to low, confirming a clear inverting action with the full-voltage swing. The average voltage gain of the inverter was estimated as 38, 50, 79, 110, and 149 at $V_{DD}$ from 10 to 50 V (Fig. 7c), respectively, which is nearly comparable to the previously reported values for oxide-based CMOS inverters[87–90]. (Supplementary Fig. 35 for statistical results on the gain of the CMOS inverter, and Supplementary Table 5 for the CMOS inverter summary) The $I_{DD}$ of the ultrathin $p$-SnO/$n$-Ga$_2$O$_3$ TFT-based CMOS inverter as a function of $V_{in}$ is shown in Fig. 7d. The static currents are lower than 1 nA, leading to the low static power dissipation of 20 nW per logic gate at a $V_{DD}$ of 50 V. The $P_{out}$ is smaller than 84 nW per logic gate, which indicates that this CMOS inverter can be operated by the nanowatt power source. (Fig. 7e) This demonstrates the high potential of atomically thin oxide-TFT-based inverter circuits for next-generation energy-efficient thin-film electronics.

The uniaxial pressure-assisted liquid-metal printing approach using nanoscale confinement growth was developed for low-temperature processed n-channel $\beta$-Ga$_2$O$_3$ TFTs for energy-efficient and cost-effective next-generation ubiquitous sustainable electronics. The presented growth method successfully fabricated device-quality crystalline $\beta$-Ga$_2$O$_3$ nanosheet at low temperatures (<150 °C) under vacuum-free, solvent-free, and non-controlled ambient air conditions. The n-channel TFTs based on $\beta$-Ga$_2$O$_3$ nanosheet exhibited high performance with a reasonably high mobility of 11.7 cm$^2$ V$^{-1}$ s$^{-1}$, an on/off-

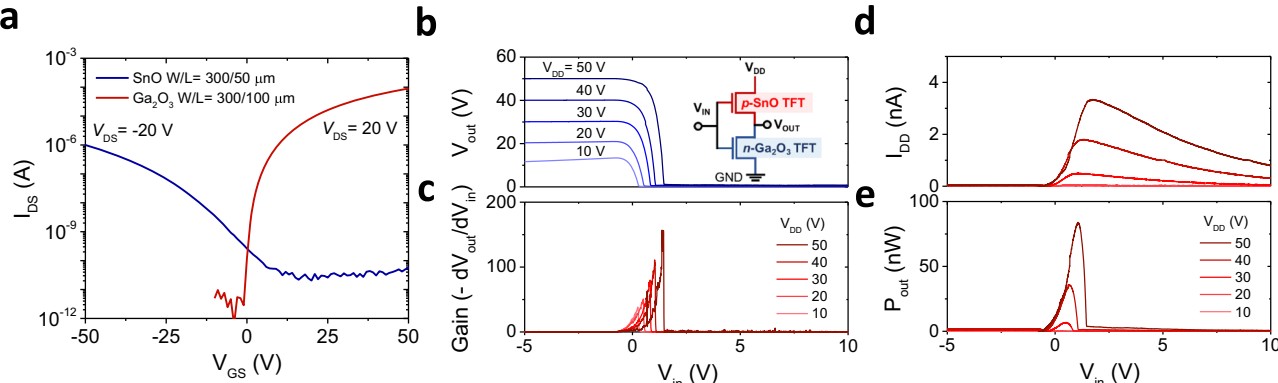

**Fig. 7 | N-Ga$_2$O$_3$/p-SnO transistor-based complementary metal-oxide-semiconductor (CMOS) inverter.** A CMOS inverter is a logic circuit consisting of a p-channel MOSFET (PMOS) as the pull-up device and an n-channel MOSFET (NMOS) as the pull-down device. **a** Typical transfer characteristics of the ultrathin p-channel SnO TFT with $V_{DS}$ = −20 V and n-channel Ga$_2$O$_3$ TFTs at $V_{DS}$ = 20 V. (The Ga$_2$O$_3$ nanosheets used for the CMOS circuits demonstration were grown using the $T_p$ of 150 °C, $P_p$ of 29 kPa.) **b** Voltage transfer characteristics (VTC). **c** Corresponding voltage gains for the p-SnO/n-Ga$_2$O$_3$ TFT-based CMOS inverter. Inset: schematic of the corresponding oxide-TFT-based CMOS inverter circuit. **d** Corresponding supply currents ($I_{DD}$). **e** Output-power consumption ($P_{out}$) for p-SnO/n-Ga$_2$O$_3$ TFT-based CMOS inverter.

current ratio of ~$10^9$, and a small subthreshold slope of 163 mV·decade$^{-1}$, which is the best device performance for low-temperature-processed Ga$_2$O$_3$ TFTs to date. We also observed p-channel operation in the off-stoichiometric GaO$_x$ channels fabricated at high-pressure conditions. Toward the oxide-based circuit application, we demonstrated low-power and all-oxide-based zero-$V_{GS}$-load NMOS and CMOS inverters using metal-liquid printing derived Ga$_2$O$_3$ nanosheet channel oxide-TFTs. These inverters showed full-voltage swing characteristics and high energy-efficient operation with low static power dissipations. Our work demonstrates the high potential of $\beta$-Ga$_2$O$_3$ for high-performance n-channel oxide TFTs and offers a promising approach for the development of sustainable oxide-TFT technology for next-generation electronics.

## Methods

### Synthesis of gallium oxide nanosheet

Atomically thin Ga$_2$O$_3$ nanosheets were fabricated by printing the oxide skin from the liquid gallium (Ga) metal. The liquid Ga metal was prepared by melting elemental Ga (99.99% purity, Shot Metals) inside a glass vial on a hot plate at 50 °C in ambient air.

To fabricate the nanosheets, a liquid droplet of the liquid Ga metal (with a size of ~1 to ~5 mm) was placed on a SiO$_2$/Si substrate using a pipette, and the substrate with liquid metal droplet was heated to the desired process temperature (80–200 °C), which is above the melting point of the liquid Ga metal. The surface of a SiO$_2$/Si substrate was treated with O$_2$ plasma (rf power = 70 W for 1 min) and preheated to the desired process temperature (80–200 °C) before a liquid Ga droplet was dropped on it. During the printing process, a second substrate (also treated with O$_2$ plasma and preheated to process temperature) was pressed onto the center of the droplet to spread the liquid alloy homogeneously between the two substrates. The two substrates were kept at process temperature (80–200 °C) and under uniaxial vertical pressure for the gallium oxide nanosheet growth for 3 min. When the printing process time was too long (>10 min), the channel exhibited insulating behavior, rendering the TFTs inoperative. (Supplementary Fig. 38) After the squeezing step with uniaxial vertical pressure, the top substrate was lifted vertically and separated without lateral slippage. Then, homogeneous ultrathin Ga$_2$O$_3$ nanosheets were exfoliated onto both substrates. The strong van der Waals bond between the oxide skin and the substrate facilitated the delamination of the oxide.

Metal inclusions attached to the exfoliated nanosheets could be removed by gently rubbing the SiO$_2$/Si wafer, which was submerged in ethanol, with a soft wiping tool (cotton bud). The Ga$_2$O$_3$ nanosheets were found to be firmly attached to the SiO$_2$ surface due to strong van der Waals adhesion between the nanosheets and the substrate, and they remained intact throughout the cleaning procedure. The large area atomically thin Ga$_2$O$_3$ nanosheets exceeding several centimeters in lateral dimensions could be fabricated efficiently using this method.

### TEM sample preparation procedure

First, the TEM grid was placed on top of the glass slide and preheated to process temperature (80–200 °C), which is above the melting point of the liquid Ga metal. Note that we did not perform O$_2$ plasma treatment for the TEM sample, as we did for the Ga$_2$O$_3$ nanosheet on the SiO2/Si substrate.

To fabricate the nanosheets, a liquid droplet of Ga metal (with a size of <1 mm) was pipetted onto a TEM grid, and the TEM grid/glass slide with the liquid metal droplet was kept heated at 50 °C to prevent the liquid Ga from solidifying. During the printing process, we used SiO$_2$/Si as the top substrate (preheated to process temperature (80–200 °C)), pressing it onto the center of the droplet to spread the liquid alloy homogeneously between the TEM grid and SiO$_2$/Si substrates. The TEM grid and SiO$_2$/Si substrates were kept at process temperature (80–200 °C) under uniaxial vertical pressure for gallium oxide nanosheet growth for 3 min.

After the squeezing step with uniaxial vertical pressure, the top SiO$_2$/Si substrate and TEM grid were carefully separated. We then used a soft wiping tool (cotton bud) to remove Ga liquid inclusions directly by gentle rubbing. It is important to note that we did not immerse the sample in ethanol, as we did for the Ga$_2$O$_3$ nanosheet on the SiO$_2$/Si substrate. Following the printing process, we observed that most carbon films were broken. During TEM analysis, we looked for nanosheets suspended and connected/supported by the bar of the TEM grid for detailed analysis. (Supplementary Fig. 13).

### Materials characterization

The surface structure of the nanosheet was analyzed using both optical microscopy and atomic force microscopy (AFM). The nanosheet thickness was determined by measuring the height of the patterned edge using AFM. The detailed crystal structures of the Ga$_2$O$_3$ nanosheets were examined using high-resolution transmission electron microscopy (HRTEM) with an acceleration voltage of 200 keV. The selected area electron diffraction (SAED) pattern was obtained using fast Fourier transform (FFT).

**Thin-film transistor fabrication and electrical characterization**

Thin-film transistors (TFTs) were fabricated with inverted-staggered structures, i.e., bottom-gate and top source/drain contacts. The 150-nm-thick thermal-oxidized $SiO_2/n^+$-Si substrates were used as the gate oxide and gate electrode. The channel area was patterned using photolithography and defined by chemical wet-etching with 1 mol/l of diluted $HNO_3$ solution. The source and drain electrodes were made of ohmic contact 40-nm-thick indium tin oxide (ITO), which was deposited by pulsed laser deposition (PLD) with an oxygen partial pressure of $2 \times 10^{-4}$ Pa at room temperature. The channel length ($L$) and the width ($W$), which are defined by metal mask processing, were 100 and 300 μm, respectively. The electrical characteristics were measured using the semiconductor parameter analyzer at room temperature in the dark.

**Device simulation**

Two-dimensional simulations of the TFTs cross-section were conducted using a 2D ATLAS TCAD simulator (Silvaco) to gain insight into the electronic and defect structures of TFTs. The simulations utilized the same configuration and materials as the TFT devices. The parameters used in the simulations are listed in Supplementary Table 3. The detailed code for the TCAD simulation is available upon request.

## Data availability

Relevant data supporting the key findings of this study are available within the article and the Supplementary Information file. All raw data generated during the current study are available from the corresponding authors upon request.

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

## Acknowledgements

This work was partly supported by faculty start-up funds at UC San Diego.

## Author contributions

K.N. conceived the concept and supervised the project. C.-H.H. designed the experiments and carried out material growth, device fabrication, and characterization. R.-H.C. and Y.-L.C. conducted material characterization, including TEM, AFM, and XPS. C.-H.H. and K.N. analyzed the results and contributed to writing the manuscript.

## Competing interests

The authors declare no competing interests
