## [Peer Review file · Nature Communications]

Low-temperature pressure-assisted liquid-metal printing for β -Ga₂O₃ thin-film transistors

Corresponding Author: Professor Kenji Nomura

Version 0:

Reviewer comments:

Reviewer #1

(Remarks to the Author)
NCOMMS-23-45192

Summary

This paper reports an interesting pressure assisted liquid metal process for printing crystalline beta-Ga₂O₃ channel materials at low temperatures (150 C) without vacuum. The performance of these devices compares favorably against solution-processed beta-Ga₂O₃, showing steep turn on and a reasonably high mobility of 11.7 cm²/Vs (though it is not clear if this performance is the as-printed film or the post-annealed film). Beyond the device results, the fundamentally interesting scientific conclusion of this work is the idea that this new kind of liquid metal printing process allows the authors to engineer a dependence of crystallinity and semiconducting character on the pressure used to post-anneal the film. This result does not appear to have been reported before in the liquid metal printed oxides community and it could be an influential idea if other authors can repeat the process reliably.

However, one on hand, it appears that the application of the pressure is more of a post-annealing step rather than printing itself. It seems that the film is held under a weight after the initial printing step. This is an important distinction because while liquid metal printing can be done in a rapid step, the annealing process is completed over a longer period of 3 minutes. This is still relatively fast compared with vacuum deposition, but it is harder to see how this process could be scalable to larger areas needed for the low-cost and flexible applications defined by the authors in the introduction.

There are a few weaknesses of this paper that need to be resolved before it can be published in Nature Comm. First, as detailed in the comments below, the device statistics are not presented clearly and there does not appear to be much information regarding whether the presented devices are from multiple substrates, multiple batches, etc. This information regarding variability should have been presented if one of the main claims of the paper is that they have developed an effective printing method. Some details of the results are obfuscated, for example, how the authors conclude the phase diagram of crystallinity as a function of pressure and temperature – were all of these individually characterized by HRTEM since there is no mention anywhere of doing measurements by XRD? Furthermore, the methods section is missing many details that would make it possible for the reader to fully reproduce parts of the work such as the TCAD simulations.

Recommendation

Reconsider after major revisions to address the deficits defined in the comments below:

Detailed Comments:

1. The authors make claims about the role of the pressure-assisted LMP process and they produce the diagram of crystallinity as a function of pressure and temperature – there is, however, no mention of doing X-ray diffraction studies on these films. Was the crystallinity of each one of these 13 films evaluated via TEM analysis? It is not necessarily clear from the figure caption or the methods section. It would be better to have XRD measurements of these films. The TEM images shown in Figure 2i are for a small area, presumably printed on a TEM grid. What is the printing pressure for the few TEM images shown? IT should be easier for the reader to find out these kind of details.

2. The use of a high pressure pressing operation to form a liquid metal printed 2D oxide film was reported last year (10.1038/s41699-022-00294-9). It would have been helpful for the authors to put their method in context against that report – is their method differentiated from the previous report of a hydraulic pressed liquid metal or is the innovation that they have discovered the dependence of the electrical properties on pressure during the initial 3 minute annealing step?

3. The surface of the beta-Ga₂O₃ films shown by AFM in Figure 2a – because there is no height map it is difficult to tell just how rough these films are.

4. There should be a high standard for device statistics for publication in Nature Comm – at minimum, I would have expected to see an explicit mention of the N for the devices shown in Figure 3. Also Figure 3 only shows a single direction curve rather than a double curve. It should show double sweeps for the transfer curves as is standard in this field. Some of the hysteresis is in fact discussed in the section around lines 370-380, but it should be shown clearly in the main figures rather than being buried in the SI.

5. The optimum device characteristics discussed in lines 365 – 380 cite device statistics, but there are not details to be found about the number of devices measured. However, there are histograms provided in Figure S4 that have data that does not match the high performance cited in the results section of the manuscript (μ is only 1-4 cm²/Vs rather than the champion performance of 11.7 cited in the abstract and elsewhere in the text). What is the the explanation for this? 1) It is pretty confusing for the reader and it makes the reviewer wonder what the actual performance is. The average in the manuscript is reported to be 8.2 for μ_{sat} while the supporting information shows it to be only 1.63 cm²/Vs. Which is it? Because the captions are not very specific, it is hard to know what is going on ..Is the higher performance after post annealing? It should be clearly stated.

-it appears that the histogram in SI must match up with the discussion on pg. 8 – are these just the non-post-annealed devices? It needs to be more clear what differentiates the various groups of device data

6. There is a claim regarding the high intrinsic gain of the devices reported in Figure S5. While the authors' results are better than most reports of thin film transistors, it is worth noting that the authors claim about being 2 orders of magnitude higher than a MOSFET is not remotely accurate. A long channel Si MOSFET would be > 10³ at comparable channel lengths. A comparison to a highly scaled ultrashort channel MOSFET is a bit meaningless here. A better comparison would be to put the intrinsic gain of their Ga₂O₃ devices up against a category of devices such as source-gated thin film transistors made from oxides.

7. Linear mobility is more reflective of the performance of this new channel material. Why is the linear mobility not reported for any of these devices? It is the standard in this field.

8. The discussion of the TCAD modeling reported on lines 428-444 does not seem to have much context here – it is not surprising for this material system that it would have this shape or magnitude of electron trap states. The authors need to do more with this data, otherwise it is hard to see what the purpose is for the fitting to produce the DOS data. Additionally, there should be greater detail regarding the device simulation. A reader would NOT be able to reproduced this work based on the details included in the paper / SI.

9. The discussion around line 500 regarding the power consumption of the inverter does not feel very substantive. What is the point of listing a power of 43 nW? this is orders of magnitude larger than any commercial technologies and there does not seem to be any reasonable point of comparison. Without having a scaled device area, it is hard to see how the power numbers are meaningful in this case. The gain of the inverter is perhaps a better metric, but the 1 kHz switching test is also not very telling. The authors are encouraged to present a test of the performance at higher frequencies that would be more relevant to any circuit level demonstration in the applications they mention in the introduction. Also, the device dimensions and gate dielectric specifications need to be more clearly mentioned to have any sense for how impressive the AC performance is. Clear benchmarking against a standard material set (sputtered IGZO used in display industry) would be a better way to approach this AC characterization. There are two figures worth of inverter characterization here, but no error bars – having a report for an average value of the gain will be important for comparing against other works in this field.

10. It is not immediately clear that the experimental setup shown in the supporting information is scalable to larger areas. The squeezing of a single droplet produces films that are only about 1 cm² in diameter. The authors need a compelling case for how this pressure assisted annealing could be scalable. Having a demonstration to show the variability across a substrate would also be important. The device statistics in this paper are relatively weak. The only histograms shown in Figure S4 have only 12 devices. Are these all from a single substrate? All of these details need to be clearly stated.

11. The other bit of context that is needed for understanding the crystallization of the beta-Ga₂O₃ phase here is a comparison to the other reports of beta-Ga₂O₃ formed by liquid metal printing (e.g. 10.1038/s41699-021-00219-y)

12. Does the printing pressure influence the thickness or uniformity of the transferred films? Is the morphology similar between the two? Microscope images showing the nanosheet morphology more in detail compared with what is currently in Figure S2 would be helpful.

13. The argument regarding the activation energy for crystallization on pg. 10 and 11 is interesting – it would have been helpful to have a sense for how the annealing time influences the crystallinity. Does a long term of applied pressure increase the crystallinity? Why was 3 minutes chosen?

Figures

1) There is no color bar to give an idea about the height map scale for parts a and b of Figure 2. The line scan itself is very small.

Additional Comments:

1) Would be good to have small English edits to details such as the figure captions

Reviewer #2

(Remarks to the Author)

The manuscript 'Low-temperature crystallization in nanoscale confinement enabled by pressure-assisted liquid-metal printing for β -Ga₂O₃ thin-film transistors' follows on from a significant published body of work in the emerging area of liquid metal printed 2D materials. The deposition of gallium oxide nanosheets from liquid metals is well known and ~30+ papers have appeared on this topic, either focusing on the direct use of the grown Ga₂O₃ or using it as a building block to make other 2D materials. Furthermore, many other 2D materials have been developed using liquid metal printing with SnO, SnO₂, ITO, and In₂O₃ showing impressive electronic device performances sometimes exceeding what has been reported in this manuscript. As such, there are some concerns reading the impact and novelty, and whether the presented work is suitable for an outlet such as Nature Communications. This is ultimately a decision for the editors.

Aside from the synthesis of Ga₂O₃, the authors report the application of mild pressure and heat that is claimed to facilitate crystallisation, leading to a variety of gallium-based oxides including beta Ga₂O₃, amorphous Ga₂O₃ and Ga₂O. Overall this aspect of the work is not convincing. The applied forces are too low to be consequential. 0.3 kgf/cm² is roughly similar to what can be applied by hand. Pressures of ~2 kgf/cm² are only slightly higher. This is almost certainly not the reason for crystallization, since printing by hand is widely used in the literature using similar pressures while leading to amorphous films.

Assuming that the materials do indeed crystallize, it is more likely that the reason triggering crystallization is a different one. Maybe the change in surface chemistry of the substrate after plasma cleaning? Or maybe another minor change to the procedure. Maybe there are minor impurities in the used materials that cause the effect? Irrespective of this, the applied energy is very unlikely to cause the crystallisation. The authors should conduct proper calculations determining the applied energy in units of J/mol and discuss if this may be a sufficient activation energy.

Furthermore, the evidence provide for the apparent crystallisation of the material is not conclusive. The authors only provide a TEM image. This is not sufficient. First of all, how did the authors prepare the TEM sample? The TEM membrane is usually soft and suspended inside a thicker copper mesh. Applying pressure from above is not likely to actually be exerted onto the Ga₂O₃ sheet. In order to convincingly show that the material has been crystallized, the authors should provide data such as:

- 1) Raman mapping
- 2) 4D STEM (or nanobeam diffraction)
- 3) Atomic resolution AFM or STM at various positions on a single sheet.

This should be conducted for a range of samples, showing that the absence of pressure leads to amorphous films, while the presence leads to crystallization. The supposed synthesis of Ga₂O (or Ga(I) oxide) has not been properly confirmed at all. This material should be thoroughly characterised since it would be a significant achievement.

The authors proposed mechanism hinges on several assumptions. One of the assumptions is that the volume of beta gallium oxide is smaller than that of the amorphous film. This is not a forgone conclusion. The Authors need to proof this. Furthermore, in order to be convincing, the authors would need to conduct DFT calculations showing that the proposed reaction is indeed feasible and can indeed be triggered by such a minimal activation energy provided by the pressure. If this was indeed the case, then crystallisation of the amorphous Ga₂O₃ phase would be easily triggered by many other stimuli including heat, which is evidently not the case (literature reports have shown heating of amorphous Ga₂O₃ to several hundred C without observing crystallisation).

Minor notes: The equation in line 153 seems to be incorrect. The free energy change (ΔG) is dependent on the change of the enthalpy (ΔE) and the change of then entropy (ΔS). The deltas have been omitted on the right-hand side of the equation.

The relevance of Figure 1 a is not clear. It might be more useful to utilize this space for something more relevant.

Reviewer #3

(Remarks to the Author)

In this work, the author introduces a pressure-assisted strategy on the reported liquid metal squeezing method to effectively induce the direct transition of 2D Ga₂O₃ from amorphous to polycrystal state at a low temperature (< 200 °C). Moreover, the pressure-assisted strategy simultaneously enables the change of electrical properties from insulator (amorphous 2D Ga₂O₃) to n-type semiconductor (polycrystalline 2D β -Ga₂O₃). The fabricated 2D β -Ga₂O₃ TFTs exhibit superior switching properties with high saturation mobility and large on/off-current ratio (~10⁹). The finding is interesting and useful for the design of high-performance oxide TFTs. However, we only have one concern that such a low pressure of 0.3 kgf/cm² can really promote the crystallization of amorphous Ga₂O₃ at a low temperature of 80 °C. Actually, TEM analysis is not adequate to support the formation of β -Ga₂O₃ and more crystal structure characterizations should be tested. In my opinion, this manuscript could not be recommended to publish before the following issues are properly addressed.

1. It is challenging to exfoliate and transfer 2D Ga₂O₃ film from substrate to TEM grid. A detailed description of the preparation/transfer process of TEM sample is easy for readers to refer and repeat. The authors should add corresponding

detailed description in the experimental section.

2. Figure 2j shows a distinct crystalline feature. However, it is only a small region with 10 nm * 10 nm. Whether 2D Ga₂O₃ amorphous film is completely transformed into polycrystal crystal should be further confirmed. SAED tests are suggested. Is it possible to exist a local region of amorphous Ga₂O₃ in the large-scale film? If yes, how does it influence the electrical performance of 2D Ga₂O₃ film? In addition, the observed crystallinity of Ga₂O₃ may come from the irradiation of electron beam which also produces heat to induce the phase transition. To fully examine the crystallization of 2D Ga₂O₃, it is necessary to conduct the HRTEM (SAED) analysis of large-area samples in short time.
3. In Figure 2i-k, the quality of TEM images is poor. It is difficult to confirm the formation of β-Ga₂O₃ from these results. A detailed and perfect TEM test should be required. Additionally, a series of XRD, Raman, XPS and PL tests are also required. It is significantly important to completely verify the formation of β-Ga₂O₃ crystal because it is the core finding of this work. Reliable experimental evidences should be provided.
4. In Figure S8a, the presented STEM image is not clear. In Figure S8d, the EDX spectrum has several un-indexed peaks. Which elements do they belong and what effect they will have to affect the electrical properties in such thin layer?
5. The author mentioned that "when the device was fabricated at 30 °C, the channel showed highly conducting behavior with an electrical conductivity of 28.3 S·m⁻¹, but the device exhibited negligible field-effect current modulation". What's the reason? Does the film have the high content of residual metallic Ga? XPS tests are suggested.
6. Actually, when we used the squeezing method to fabricate 2D Ga₂O₃, it also exist an assisted pressure approaching to 0.3 kgf/cm². However, the fabricated 2D Ga₂O₃ film is insulated. So, what's the key point and technology of your proposed PA-LMP method to induce the crystallization of such thin 2D-Ga₂O₃ layer? How about the repeatability or success rate of the experiment in Ga₂O₃ layer crystallization and device? It is suggested to provide a video of preparation process of PA-LMP.
7. Have the authors tried higher assisted pressure than 1.3 kgf/cm² and higher post-annealing temperature than 400 °C in the preparation process? How it will affect the crystal quality and electrical performance of 2D Ga₂O₃ film?
8. How about the continuity of the film? Does it exist cracks and holes? Please give low-magnification optical image and more description in the main text.
9. Are there any tiny liquid Ga droplets (normally < 5nm) on the surface of 2D Ga₂O₃ film? High-resolution SEM observation is required to check it.
10. Why the authors choose ITO as the materials of source and drain electrodes? Have you tried Ti/Au metal electrodes? Does it influence the performance of TFTs?
11. The optical photograph and SEM images of TFTs should be added in the supporting information.
12. How about the stability of 2D Ga₂O₃ film and related TFTs performance in the ambient air atmosphere after one week or one month?
13. How about the photoresponse of 2D β-Ga₂O₃ film? It is also an important property of 2D Ga₂O₃ film. Does it only respond to deep ultraviolet light?

Version 1:

Reviewer comments:

Reviewer #1

(Remarks to the Author)

The authors have addressed the majority of the comments from the first round of review.

I have a few other points that need resolution:

- 1) There is no reasonable reason to use kgf as the units for the pressure. Just use Pa / kPa, etc. Kgf will confuse readers unnecessarily since it is a non-standard unit
- 2) The low power consumption is really not a unique advantage of this material compared to any other standard TFT technology. The authors severely overstate the capabilities in terms of low power consumption. Comparing it against silicon CMOS is not a good way to go – standard MOSFET devices could easily be made with low power consumption. The existence of the entire field of low power VLSI systems alone should prove that the authors argument is a bit of a reach here. I would like to see removal or at the very least the tempering of the claims to low power logic operation. The devices are good but not necessarily unique compared to the state of the art in metal oxide TFTs and other low power devices. The discussion on pg. 25 should be updated to reflect this. All that being said, the authors do certainly show low off-state currents. This is a good thing. The authors should have written the discussion of this fact to cite the literature on IGZO displays, which highlights their off currents as an advantage. I recommend doing this rather than trying to rationalize whether 84 nW is a significantly lower power consumption.
- 3) On pg. 13 the authors mention that the density of their amorphous GaO_x was observed to be lower?? Did they measure it with x-ray methods or are they citing the literature shown there? It is unclear and misleading
- 4) Do the authors have any evidence that the gap at these ranges of pressure is at a sufficiently small length scale to result in the confined crystallization hypothesized? I would want to see some additional evidence of this claim before they cite the SI discussion on 2D confined crystallization.

There are still a number of minor grammatical issues to resolve. Here are a couple of examples. These need to be fixed.

“The static currents are lower than 1 nA, leading to the low static power dissipation is 20 nW per logic gate at VDD of 50V”

The low-power oxide-CMOS inverter using p-SnO/n-Ga₂O₃ TFTs is attributed to the ultralow off-current nature for both the n- and p-channel transistors.

Reviewer #2

(Remarks to the Author)

While the authors have conducted a lot of extra work, they failed to genuinely address and engage with the core criticism that was raised. A pressure of 1.3 kgf/cm² (or ~1.3 bar) is inconsequential. There is just not enough energy applied to impact crystallization. Let's be clear, I believe that the authors made high performing devices from LM printed Ga₂O₃. I might also be convinced that the better performance is linked to changes in crystallinity. I just don't believe that the applied pressure has anything to do with the change in performance / crystallinity. Something else must be going on that impacts crystal growth / electronic behaviour. If the paper is published as is there is a high chance that the core mechanism that is put forward is incorrect. As such, the authors should really put forward a strong case regarding how such small pressure can have an effect. This needs to go beyond showing that there are changes in crystallinity. It needs to link these changes to the tiny increase in pressure.

Aside from this, the authors also did not show that the material is actually crystalline - Yes, they show crystalline domains but there is no evidence that the material is crystalline beyond the scale of the TEM images - i.e. ~10 to 20 nm. The material may be amorphous with occasional crystalline inclusions/domains.

Raman mapping would go a long way. Alternatively a systematic study on a larger sample of 2D Ga₂O₃ in the TEM with many sections being measured at different locations would help. This would have to be done for both, the crystalline sample prepared under pressure and the amorphous sample prepared without pressure. I do acknowledge that the cross sectional samples are moving towards this, but as it stands these would need to be supplemented.

The GIXRD measurements are not that useful at this point. As the authors point out, getting x-ray diffraction data on thin films is extremely difficult. The indicated peaks are as far as I can tell in the noise level and cannot be identified with confidence. It is also not possible to tell if these peaks (if they are there) arise from the 2D film or from thick inclusions.

I do want to highlight that the work on the electronic device aspects has been significantly strengthened and expanded. These aspects are definitely worthy of publication. The synthesis aspects and the proposed mechanism are the issue.

Reviewer #3

(Remarks to the Author)

The revisions can address our concerns in full and the revised manuscript is suitable for publication.

We still suggest the author to provide a video of preparation process of PA-LMP. It is easier for the readers to repeat the experiments.

Version 2:

Reviewer comments:

Reviewer #1

(Remarks to the Author)

The revision properly addresses each point from the previous round of review. The manuscript is ready for publication.

Reviewer #2

(Remarks to the Author)

Once again I feel that the authors have not genuinely attempted to show that such a low pressure can indeed cause crystallisation. To be clear - I do not doubt that crystallisation occurs. 129 kpa is just not enough. For comparison, this pressure is actually on the lower end of what is found in soda cans. It is just not convincing that this low pressure can cause changes in crystallisation habit. The chemistry does not make sense. When we conduct similar experiments in our lab and apply pressure to the 2D GaOx film we do not see the reported crystallisation, however we did not spend as much time on this as the authors have.

Having said the above. I feel that after 2 years of revisions the editor needs to make a final call. The authors are clearly convinced that they got it right. I don't think that I can be convinced. Maybe it is time to just publish the work and if people can reproduce it then I am happy to be proven wrong. Maybe in a future paper a different mechanism is proposed that is more viable. Or, the paper is found to not be reproducible and the field moves on.

Response to the reviewer's comments and the list of corrections

Manuscript ID: NCOMMS-23-45192

Title: Low-temperature crystallization enabled by pressure-assisted liquid-metal printing for β -Ga₂O₃ thin-film transistors

Authors: Chi-Hsin Huang, Ruei-Hong Cyu, Yu-Lun Chueh and Kenji Nomura*

Reviewer #1 (Remarks to the Author)

This paper reports an interesting pressure assisted liquid metal process for printing crystalline beta-Ga₂O₃ channel materials at low temperatures (150 C) without vacuum. The performance of these devices compares favorably against solution-processed beta-Ga₂O₃, showing steep turn on and a reasonably high mobility of 11.7 cm²/Vs (though it is not clear if this performance is the as-printed film or the post-annealed film). Beyond the device results, the fundamentally interesting scientific conclusion of this work is the idea that this new kind of liquid metal printing process allows the authors to engineer a dependence of crystallinity and semiconducting character on the pressure used to post-anneal the film. This result does not appear to have been reported before in the liquid metal printed oxides community and it could be an influential idea if other authors can repeat the process reliably.

However, one on hand, it appears that the application of the pressure is more of a post-annealing step rather than printing itself. It seems that the film is held under a weight after the initial printing step. This is an important distinction because while liquid metal printing can be done in a rapid step, the annealing process is completed over a longer period of 3 minutes. This is still relatively fast compared with vacuum deposition, but it is harder to see how this process could be scalable to larger areas needed for the low-cost and flexible applications defined by the authors in the introduction.

There are a few weaknesses of this paper that need to be resolved before it can be published in Nature Comm. First, as detailed in the comments below, the device statistics are not presented clearly and there does not appear to be much information regarding whether the presented devices are from multiple substrates, multiple batches, etc. This information regarding variability should have been presented if one of the main claims of the paper is that they have developed an effective printing method. Some details of the results are obfuscated, for example, how the authors conclude the phase diagram of crystallinity as a function of pressure and temperature – were all of these

individually characterized by HRTEM since there is no mention anywhere of doing measurements by XRD? Furthermore, the methods section is missing many details that would make it possible for the reader to fully reproduce parts of the work such as the TCAD simulations.

(Response) Thank you for the reviewer's thoughtful comments and constructive suggestions. We appreciate the opportunity to address their concerns and provide point-to-point responses with additional experimental data to further support our findings.

(Detailed Comments)

Comment 1. *The authors make claims about the role of the pressure-assisted LMP process and they produce the diagram of crystallinity as a function of pressure and temperature – there is, however, no mention of doing X-ray diffraction studies on these films. Was the crystallinity of each one of these 13 films evaluated via TEM analysis? It is not necessarily clear from the figure caption or the methods section. It would be better to have XRD measurements of these films. The TEM images shown in Figure 2i are for a small area, presumably printed on a TEM grid. What is the printing pressure for the few TEM images shown? IT should be easier for the reader to find out these kind of details.*

(Response) Following the reviewer's suggestion, we further conducted TEM analysis for the Ga₂O₃ nanosheet prepared by a pressure-assisted liquid metal printing (PALMP) with different process conditions. (*i.e.* printing temperature (T_p) and process pressure (P_p)) to characterize the crystallinity of β-Ga₂O₃. **Figure R1-R3** presents TEM analysis, including low and high-magnified TEM images and selected area electron diffraction (SAED) pattern. (**Figure R1** for the T_p of 150 °C and the P_p of 1.3 kgf/cm², **Figure R2** for 150 °C and 0.3 kgf/cm², and **Figure R3** for 80 °C and 0.3 kgf/cm². Respectively.) All of these TEM analyses confirm that the Ga₂O₃ nanosheet grown under these process parameters is β-Ga₂O₃. Additionally, the TEM analysis found that high-temperature and high-pressure conditions (*i.e.* T_p of 150 °C and P_p of 1.3 kgf/cm² (**Figure R1**)) produce β-Ga₂O₃ nanosheet with better crystallinity. The observation is directly connected to the reason why the β-Ga₂O₃-TFT fabricated at high-temperature and high-pressure conditions exhibited improved device performances, which operated with high mobility in the range of 8~10 cm²V⁻¹s⁻¹.

Following the reviewer's concerns regarding the analytic area of TEM, we also conducted a wide area of cross-sectional TEM analysis for the Ga₂O₃ nanosheets grown

at the T_p of 150 °C and the P_p 1.3 kgf/cm². (**Figure R4**) The corresponding selected area electron diffraction (SAED) exhibited a clear spot pattern, confirming that the nanosheet is composed of polycrystalline. However, we consider it challenging to perform a reliable analysis to identify the crystal phase of the material in such ultra-thin thickness samples. Thus, we newly prepared a thick-Ga₂O₃ nanosheet with a thickness of approximately 10 nm by a triple-printing process to conduct a reliable SAED analysis. **Figure 5** summarizes cross-sectional TEM analysis for the thick-Ga₂O₃ nanosheet. ((a) for a low-magnification TEM image with SAED measurement area and (b) the corresponding SAED patterns). All the SAED patterns showed distinct spot patterns that can be indexed to the crystal plane of β -Ga₂O₃ phase. These results concluded that the developed Ga₂O₃ nanosheet grown by PA-LMP is a crystalline β -Ga₂O₃ phase over a large area.

Following the suggestion provided by the reviewer, moreover, we performed grazing incidence X-ray diffraction (GIXRD) analysis for the Ga₂O₃ nanosheet prepared by different printing conditions and summarized these XRD patterns in **Figure R6** (**green line**: LMP at the T_p of 150 °C without pressure, **blue line**: the P_p of 0.3 kgf/cm² and the T_p of 80 °C, **red line**: 0.3 kgf/cm² and 150 °C, **purple line**: 1.3 kgf/cm² and 150 °C, respectively). The powder diffraction file (PDF #01-087-1901) was used to assign the diffraction peaks for β -Ga₂O₃. A halo peak observed around $\sim 24^\circ$ is attributed to the glass substrate. No diffraction peaks were observed for the Ga₂O₃ prepared by the LMP method without applied pressure (**green line**), indicating that the nanosheet was amorphous. On the other hand, the Ga₂O₃ nanosheet grown by the PA-LMP exhibited a series of diffraction peaks assigned to the β -Ga₂O₃ crystal phase. (Note: the intensity of the diffraction peaks is weak due to the thin thickness of Ga₂O₃ being ~ 3 nm.)

Furthermore, we also performed TEM analysis for the GaO_x nanosheets grown by PA-LMP at the lowest temperature and applied pressure. (i.e., the T_p of 30°C and the P_p of 0.3 kgf/cm² (**Figure R7**)) From the SAED pattern, we confirmed that the nanosheet was amorphous (*i.e.* a-GaO_x). (**Figure R7(b)**) Additionally, we observed the presence of residual metallic Ga in the a-GaO_x nanosheets. (**Figure R7(c)**) The HRTEM image showed the crystal lattice structure with internal spacings of ~ 0.276 nm, which are assigned to the (333) plane of the crystal structure of γ -Ga metal. (**Figure R7(d)**) **Figure R8** shows the energy dispersive X-ray spectroscopy (EDX) analysis for the a-GaO_x nanosheets. ((a) Ga and O elemental mapping and (b) EDX spectra). The EDX analysis found the atomic ratio of the a-GaO_x nanosheet of Ga:O = 58.2:41.8%, indicating that the nanosheet is Ga-rich. (**Figure R8(d)**)

(Revised parts in the manuscript)

(1) We have incorporated the additional data of **Figure R1** into Figure 2 in the main text of the revised manuscript. Additionally, **Figures R1, R2, and R3** have been added as **Figures S15, S16, and S17** in the supporting information. The revised sentence in the manuscript, “In contrast, a distinct polycrystalline feature, i.e., ordered atomic alignment, for the PA-LMP grown nanosheet (T_p of 150 °C and P_p of 1.3 kgf/cm²) was observed in the TEM analysis (**Figure 2(i)**, **Figure S15** for the low-magnified HRTEM images of the Ga₂O₃ nanosheet). The HRTEM image also showed the crystal lattice structure with internal spacings of ~0.362 nm ~0.263 nm, which are assigned to the (201) and (–111) planes, respectively, of the monoclinic β -Ga₂O₃ crystal structure. (**Figure 2(j)**) The corresponding selected area electron diffraction (SAED) pattern is shown in **Figure 2(k)**, and exhibits spots indexed to the (–111), (201), (–311), and (400) crystal plans. The observation concluded that the presented nanosheet was randomly oriented β -Ga₂O₃ crystals. In addition, we performed TEM analysis for the Ga₂O₃ nanosheet prepared using the PA-LMP approach with different process parameters to confirm the direct growth of β -Ga₂O₃ crystals. (**Figure S15-17**, Supporting Information)”

“In contrast, the PA-LMP directly synthesized a semiconducting polycrystalline β -Ga₂O₃ channel in the range of the T_p of 80–200 °C with the P_p of 0.3 kgf/cm² and in the range of the T_p 150–200 °C with the P_p of 1.3 kgf/cm². The polycrystalline nature of β -Ga₂O₃ is confirmed by both GIXRD and TEM characterization. (**Figure S15, S16, S17 and S19**, Supporting Information).”

(2) **Figures R4, and R5** have been included as **Figures S20, and S21** in the supporting information. The revised sentence in the manuscript: “Furthermore, we conducted a cross-sectional TEM analysis for the Ga₂O₃ nanosheet prepared using the liquid metal printing approach on the Si/SiO₂. The Ga₂O₃ nanosheet was prepared at the T_p of 150 °C, and the P_p of 1.3 kgf/cm², exhibiting a high TFT mobility of 8~10 cm²V⁻¹s⁻¹. The distinct polycrystalline nature of β -Ga₂O₃ was also observed, providing direct evidence that liquid metal printing can grow crystalline β -Ga₂O₃ on the SiO₂/Si substrate. (**Figure S20, S21**, Supporting Information)”

(3) The GIXRD data (**Figures R6**) have been included in the supporting information. (**Figure S19**) The revised sentence in the manuscript: “The grazing incidence X-ray diffraction (GIXRD) analysis also supported that the Ga₂O₃ nanosheets prepared by the PA-LMP method were polycrystalline β -Ga₂O₃. (**Figure S19**, Supporting Information)”

(4) **Figures R7 and R8** have also been added to the supporting information (as **Figures S25, and S26**). The revised sentence in the manuscript, “From the TEM characterization, it was also confirmed that Ga-rich GaO_x with embedded Ga metal in nanosheets grown by the printing process at the T_p of 30°C and the P_p of 0.3 kgf/cm². (**Figure S25, 26**, Supporting Information)”

-For the Reviewer’s comment, “It is not necessarily clear from the figure caption or the methods section.”, We have improved the figure captions and the methods section.

Figure R1. TEM characterization of the Ga₂O₃ nanosheet synthesized by the PA-LMP route with the T_p of 150 °C and the P_p of 1.3 kgf/cm². (a) Low-magnified HRTEM images of the Ga₂O₃ nanosheet. (b) and (d) Corresponding high-magnified HRTEM images and (c) SAED pattern.

Figure R2. TEM characterization of Ga₂O₃ nanosheet synthesized by the PA-LMP route with the T_p of 150 °C and the P_p of 0.3 kgf/cm². (a) Low-magnified HRTEM images of Ga₂O₃ nanosheet. (b), (d) Corresponding high-magnified HRTEM image and (c) SAED pattern.

Figure R3. TEM characterization of Ga_2O_3 nanosheet synthesized by the PA-LMP route with the T_p of $80\text{ }^\circ\text{C}$ and the P_p of 0.3 kgf/cm^2 . (a) Low-magnified HRTEM images of Ga_2O_3 nanosheet. (b),(d) Corresponding high-magnified HRTEM image and (c) SAED pattern.

Figure R4. (a) Cross-sectional TEM characterization of Ga_2O_3 nanosheet synthesized by the PA-LMP route with the T_p of $150\text{ }^\circ\text{C}$ and the P_p of 1.3 kgf/cm^2 . (b) Corresponding cross-sectional HRTEM image and (c) corresponding diffraction pattern extracted by the fast Fourier transform.

Figure R5. (a) Cross-sectional TEM characterization of a thick Ga_2O_3 nanosheet (the thickness of approximately 10 nm) synthesized by the PA-LMP with the T_p of 150 °C and the P_p of 1.3 kgf/cm². (b) Corresponding SAED pattern from rectangle areas (1), (2), (3), and (4), respectively.

Figure R6. GIXRD patterns for typical Ga₂O₃ nanosheet grown by PA-LMP method with different process parameters. (**blue line:** T_p of 80 °C and the P_p of 0.3 kgf/cm², **red line:** T_p of 150 °C and the P_p of 0.3 kgf/cm², **purple line:** T_p of 150 °C and the P_p of 1.3 kgf/cm²). For the comparison, Ga₂O₃ nanosheet grown by the LMP at the T_p of 150 °C without pressure was shown as **green line**. The powder diffraction file (PDF #01-087-1901) was used to identify the diffraction peaks. A halo peak around ~24° is due to the glass substrate.

Figure R7. TEM characterization of GaO_x nanosheet synthesized by the PA-LMP route with T_p of 30 °C and P_p of 0.3 kgf/cm². (a) Low-magnified HRTEM images of GaO_x nanosheet. (b) Corresponding SAED pattern. (c) HRTEM image for the Ga metal region (d) Corresponding high-magnified HRTEM image and diffraction pattern extracted by the fast Fourier transform. (inset)

Figure R8. (a) Scanning transmission electron microscopy (STEM) image of GaO_x nanosheet (T_p of 30 °C and P_p of 0.3 kgf/cm²) and the corresponding EDS composition mappings of (b) Ga and (c) O elements. (d) Corresponding EDX spectrum for the GaO_x nanosheet. The inset shows the atomic ratio of the Ga₂O₃ nanosheet of Ga:O = 58.2 : 41.8 %.

Comment 2. *The use of a high pressure pressing operation to form a liquid metal printed 2D oxide film was reported last year (10.1038/s41699-022-00294-9). It would have been helpful for the authors to put their method in context against that report – is their method differentiated from the previous report of a hydraulic pressed liquid metal or is the innovation that they have discovered the dependence of the electrical properties on pressure during the initial 3 minute annealing step?*

(Response) Thank you for recommending the reference and providing this helpful suggestion. Our method is close to the LMP approach proposed in the paper (10.1038/s41699-022-00294-9).¹ However, we innovatively applied this approach with applied pressure to grow semiconducting β -Ga₂O₃ for electronic device application.

This literature provides insightful information regarding the unique properties of the LMP method, which can directly grow crystalline oxide conductors, avoiding insulating intermediate phases and eliminating the thermodynamic barriers posed by precursor decomposition.¹ Interestingly, they presented better device performances of the 2D-In₂O₃-TFTs than the conventional In₂O₃-TFTs fabricated by PVD/ALD methods due to the effect of the overlapping crystal grain structure. We also suspected that it is due to the liquid metal printing that it is naturally easier to grow high-quality oxide materials.

Furthermore, we also found in other works (including our previous research) that the LMP method can grow oxide materials exhibiting high electron mobility at relatively low process temperatures compared to materials fabricated by conventional thin-film deposition processes.¹⁻³ Therefore, we also believe that the oxidation process of the liquid metal surface has a different thermodynamic barrier and path to grow oxide materials, resulting in it being easier to grow high-quality oxide materials at low process temperatures.

Given that the thermodynamic barrier and pathway may differ when employing a liquid metal oxidation process, the crystallization of the β -Ga₂O₃ could be easily induced by various stimuli, such as heat, pressure, etc. For example, the other literature (DOI: 10.1038/s41699-021-00219-y) also demonstrates the LMP approach by scraping off the parent metal layer with stress at a process temperature of 200°C, successfully synthesizing crystalline β -Ga₂O₃.⁴

Following the review’s suggestion, the revised section in the manuscript: “The LMP method is proposed for forming high-quality crystalline oxide materials with superior electrical properties compared to conventional physical/chemical vapor thin-film

deposition, owing to its distinct thermodynamic pathway. Therefore, LMP method has already been demonstrated to develop high-mobility Indium-based *n*-channel oxide TFTs. Here, the PA-LMP approach, which involves applying external uniaxial pressure during oxide skin formation on liquid metal, is firstly demonstrated to directly form high-quality polycrystalline β -Ga₂O₃ nanosheets at a low temperature of 150°C under a non-controlled ambient air atmosphere, exhibiting excellent transistor operation.”

Comment 3. *The surface of the beta-Ga2O3 films shown by AFM in Figure 2a – because there is no height map it is difficult to tell just how rough these films are.*

(Response) We have added the corresponding height map to the AFM images in **Figure 2(a), (b)** in the revised manuscript. **(Figure R9)**

Figure R9. AFM image and the cross-sectional step-height profile for the Ga₂O₃ nanosheet were prepared by (a) conventional LMP (without applied process pressure) and (b) PA-LMP methods (applied process pressure: 1.3 kgf/cm²).

Comment 4. *There should be a high standard for device statistics for publication in Nature Comm – at minimum, I would have expected to see an explicit mention of the N for the devices shown in Figure 3. Also Figure 3 only shows a single direction curve rather than a double curve. It should show double sweeps for the transfer curves as is standard in this field. Some of the hysteresis is in fact discussed in the section around lines 370-380, but it should be shown clearly in the main figures rather than being buried in the SI.*

(Response) We have added the device numbers in the revised manuscript and figure caption for **Figure 3**. (**Figure R10**) Additionally, we have updated all the transfer characteristics in the main figure to include the double sweep I - V curve. (**Figure R10**)

The revised section in the manuscript: “The data are analyzed using 16 representative working devices from different samples.”

Figure R10. (a) Variation of transfer characteristics for the Ga₂O₃ nanosheet TFTs with different printing process temperatures under the uniaxial process pressure of 0.3 kgf/cm². The corresponding TFT parameters (b) μ_{sat} and μ_{lin} , (c) s -value and D_{it} , (d) V_{th} , and (e) on/off current ratio are plotted. (f) Variation of transfer characteristics for the Ga₂O₃ TFTs with different process pressures under the printing process temperature of 150 °C. Corresponding TFT parameters (g) μ_{sat} and μ_{lin} , (h) s -value and D_{it} , (i) V_{th} , and (j) on/off current ratio are plotted. (The data are calculated using 16 representative working devices from different samples.)

Comment 5. *The optimum device characteristics discussed in lines 365 – 380 cite device statistics, but there are not details to be found about the number of devices measured. However, there are histograms provided in Figure S4 that have data that does not match the high performance cited in the results μ section of the manuscript (μ is only 1-4 cm²/Vs rather than the champion performance of 11.7 cited in the abstract*

and elsewhere in the text). What is the the explanation for this? 1) It is pretty confusing for the reader and it makes the reviewer wonder what the actual performance is. The average in the manuscript is reported to be 8.2 for μ_{sat} while the supporting information shows it to be only 1.63 cm^2/Vs . Which is it? Because the captions are not very specific, it is hard to know what is going on. Is the higher performance after post annealing? It should be clearly stated. -it appears that the histogram in SI must match up with the discussion on pg. 8 – are these just the non-post-annealed devices? It needs to be more clear what differentiates the various groups of device data

(Response) Thank you for carefully reviewing our manuscript. We apologize for any confusion in our original version. In the initial version of the manuscript, we presented the different electrical characteristics of the Ga_2O_3 nanosheets grown using conventional LMP and PA-LMP methods in **Figure 2**. The device fabricated by PA-LMP with the T_p of 80°C and the P_p of $0.3 \text{ kgf}/\text{cm}^2$ showed an average mobility of $\sim 1.63 \text{ cm}^2\text{V}^{-1}\text{s}^{-1}$. The histograms of device performances obtained from the device-to-device statistical analysis was provided in **Figure S4**. In **Figure 3**, we further investigated the effects of printing process pressure and temperature, achieving improved TFT mobility with a high μ_{sat} of $11.7 \text{ cm}^2\text{V}^{-1}\text{s}^{-1}$ when the Ga_2O_3 nanosheets was grown at T_p of 150°C and P_p of $1.3 \text{ kgf}/\text{cm}^2$.

To avoid confusion in the revised manuscript, we directly present device data and TEM fabricated by PA-LMP with a high P_p of $1.3 \text{ kgf}/\text{cm}^2$ and the T_p of 150°C , demonstrating high mobility in the range of $8\sim 10 \text{ cm}^2\text{V}^{-1}\text{s}^{-1}$. (Revised **Figure 2**) We then discuss the effects of process pressure and temperature in **Figure 3**. Additionally, we include the device data and relevant discussion regarding the device-to-device statistical analysis within a single chip sample (under both low and high-pressure conditions) and the batch-to-batch data for devices fabricated under low pressure conditions in the PA-LMP. (**Figure S7, Figure S8, and Figure S9**) We have provided the response regarding the uniformity of the TFT characteristics in comment 10 of Reviewer 1. Please also refer to our response for detailed information.

Comment 6. *There is a claim regarding the high intrinsic gain of the devices reported in Figure S5. While the authors' results are better than most reports of thin film transistors, it is worth noting that the authors claim about being 2 orders of magnitude higher than a MOSFET is not remotely accurate. A long channel Si MOSFET would be $> 10^3$ at comparable channel lengths. A comparison to a highly scaled ultrashort channel MOSFET is a bit meaningless here. A better comparison would be to put the*

intrinsic gain of their Ga₂O₃ devices up against a category of devices such as source-gated thin film transistors made from oxides.

(Response) In response to the reviewer’s comment, we have revised the comparison and included the performances for traditional ohmic contact a-IGZO-TFT and source-gate oxide TFT (Schottky-barrier oxide TFTs)⁵⁻⁷. **(Figure R11)**,

We added relevant references and revised the following paragraph in the manuscript, “The intrinsic gain, defined as $A_i = g_m/g_d$, was as high as 1,000 at V_{DS} of 30 V, significantly higher than that for other source-gate oxide TFTs (Schottky-barrier oxide TFTs) and one order of magnitude higher than that for conventional ohmic contact a-IGZO-TFTs. **(Figure S12 (c), Supporting Information)**”

Figure R11. (a) Output characteristics for the β -Ga₂O₃ nanosheet TFT (T_p of 150 °C and P_p of 1.3 kgf/cm²). (b) extracted $1/g_d$ (blue circles) and g_m (red squares) as a function of V_{GS} at $V_{DS} = 30$ V. (c) Intrinsic gain, g_m/g_d , as a function of V_{GS} . The intrinsic gain for ohmic contact IGZO TFTs and source-gate oxide TFTs (Schottky-barrier oxide TFTs) are also shown for comparison.

Comment 7. Linear mobility is more reflective of the performance of this new channel material. Why is the linear mobility not reported for any of these devices? It is the standard in this field.

(Response) Thank you for the insightful comments and suggestions. We have provided the linear mobility in the revised manuscript and incorporated the linear mobility data into **Figure 3** in the revised manuscript. Additionally, we have included the linear transfer characteristic **(Figure R12)** in the supporting information. **(Figure S24)**

Figure R12. Transfer characteristics for linear ($V_{DS}= 1\text{V}$) and saturation region ($V_{DS}= 20\text{V}$) of the Ga_2O_3 nanosheet TFTs with different conditions: (a) T_p of $80\text{ }^\circ\text{C}$ and P_p of 0.3 kgf/cm^2 . (b) $150\text{ }^\circ\text{C}$ and 0.3 kgf/cm^2 . (c) $150\text{ }^\circ\text{C}$ and 1.3 kgf/cm^2 , respectively.

Comment 8. *The discussion of the TCAD modeling reported on lines 428-444 does not seem to have much context here – it is not surprising for this material system that it would have this shape or magnitude of electron trap states. The authors need to do more with this data, otherwise it is hard to see what the purpose is for the fitting to produce the DOS data. Additionally, there should be greater detail regarding the device simulation. A reader would NOT be able to reproduced this work based on the details included in the paper/SI.*

(Response) We have made a more detailed discussion and information of the TCAD modeling in the revised manuscript and table in supporting information for readers to reproduce the TCAD simulation. In addition, the detailed code for the TCAD simulation is available upon request from readers.

Therefore, we added the revised paragraph in the TCAD modeling section in the revised manuscript: “To gain further insight into the variation of TFT characteristics, we performed the Technology Computer-Aided Design (TCAD) device simulation to extract the subgap defect density of state. **Figure 5 (a)** shows the measured and simulated transfer curves of Ga_2O_3 TFTs with different process conditions. Only parameter optimization for the subgap acceptor-like defect density of state (DOS) can reproduce the measured transfer curves, indicating that the variation in TFT characteristics under different process parameters mainly originates from the change in subgap defect DOS in the Ga_2O_3 channels. The extracted subgap defect DOS near the conduction band (CB) of Ga_2O_3 TFTs under different process conditions are shown in

Figure 5 (b). (Table S3 provides the parameters of the TFT simulations). We found that all the measured I - V curves were reproduced by only optimizing the acceptor-like defect (i.e., electron trap defect) with Gaussian distribution type,

$$g_G(E) = N_{GD} \cdot \exp \left\{ - \left[\frac{(E_c - E_{GA})}{W_{GD}} \right]^2 \right\}$$

where N_{GD} is the state densities at the central energy E_{GA} of the Gaussian distribution, E_c is the conduction band edge energy for the reference zero point, E_{GA} is the central energy of $g_G(E)$, W_{GD} is the characteristic decay energy. Due to the polycrystalline nature of the Ga₂O₃, we opted not to employ the acceptor-like exponential DOS to fit the measured I - V curves that are used for the tail states near the conduction for the amorphous silicon (a-Si:H) and amorphous IGZO.⁸ All the devices show a low carrier concentration of $5.5 \times 10^{14} \text{ cm}^{-3}$ due to the ultra-wide gap nature of Ga₂O₃.

We observed shallow acceptor-like defect states located at 0.15 eV below the conduction band ($E_c - 0.15 \text{ eV}$) in the Ga₂O₃ TFTs. These acceptor-like defect states primarily function as electron traps in the n -channel oxide TFT devices. The physical origin of these shallow acceptor-like defect states remains elusive; however, we suspect these defects result from Ga vacancies or weakly bonded (excess) oxygen defects.⁹⁻¹¹ The Ga₂O₃ TFT fabricated with the low-pressure condition (T_p of 80 °C, P_p of 0.3 kgf/cm²) exhibited a relatively high density of shallow acceptor-like defect states of $2 \times 10^{18} \text{ cm}^{-3} \text{ eV}^{-1}$ and remained unchanged even in the device fabricated at 150 °C. (150 °C, 0.3 kgf/cm²) On the other hand, we found that the high-pressure condition (150 °C, 1.3 kgf/cm²) effectively reduced the shallow subgap acceptor-like defect DOS to $5 \times 10^{17} \text{ cm}^{-3} \text{ eV}^{-1}$, resulting in better TFT performances with higher mobility. This acceptor-like defect reduction makes moving the Fermi level toward the mobility edge easier, achieving band-like conduction and explaining the higher mobilities observed in these TFTs.

We also found that the shallow subgap defect DOS significantly impacts the on-current and s -value but does not affect the turn-on voltage, resulting from the enhancement mode operation. The D_{it} , estimated from the s -values of the experimental transfer characteristics, are $5.37 \times 10^{11} \text{ cm}^{-2} \text{ eV}^{-1}$ (80 °C, 0.3 kgf/cm²), $4.78 \times 10^{11} \text{ cm}^{-2} \text{ eV}^{-1}$ (150 °C, 0.3 kgf/cm²) and $1.93 \times 10^{11} \text{ cm}^{-2} \text{ eV}^{-1}$ (150 °C, 1.3 kgf/cm²) for these three process conditions, respectively. The shallow subgap defect DOS for all these three conditions we attained from the TCAD simulation correspond to area densities of $6 \times 10^{11} \text{ cm}^{-2} \text{ eV}^{-1}$ (80 °C, 0.3 kgf/cm²), $4.5 \times 10^{11} \text{ cm}^{-2} \text{ eV}^{-1}$ (150 °C, 0.3 kgf/cm²) and $1.8 \times 10^{11} \text{ cm}^{-2} \text{ eV}^{-1}$ (150 °C, 1.3 kgf/cm²), respectively, considering the Ga₂O₃ thickness of 3 nm. These values are consistent with those calculated from experimental s -values.”

Also, we added this statement in the method section of the revised manuscript to mention that the code for the TCAD simulation is available for readers: “The detailed code for the TCAD simulation is available upon request.”

Comment 9. *The discussion around line 500 regarding the power consumption of the inverter does not feel very substantive. What is the point of listing a power of 43 nW? this is orders of magnitude larger than any commercial technologies and there does not seem to be any reasonable point of comparison. Without having a scaled device area, it is hard to see how the power numbers are meaningful in this case. The gain of the inverter is perhaps a better metric, but the 1 kHz switching test is also not very telling. The authors are encouraged to present a test of the performance at higher frequencies that would be more relevant to any circuit level demonstration in the applications they mention in the introduction. Also, the device dimensions and gate dielectric specifications need to be more clearly mentioned to have any sense for how impressive the AC performance is. Clear benchmarking against a standard material set (sputtered IGZO used in display industry) would be a better way to approach this AC characterization. There are two figures worth of inverter characterization here, but no error bars – having a report for an average value of the gain will be important for comparing against other works in this field.*

(Response) *1. power consumption of the inverter:* As the reviewer pointed out, modern low-power technology reduces power consumption through scaled devices and high-k gate dielectric materials. However, the presented device in this work employs a thick (the thickness of 150 nm) SiO₂ gate dielectric and a long channel length of 100 μm, resulting in a large V_{DD} in inverter operation. We acknowledge that the reviewer’s comment regarding power consumption is not applicable to the presented device. Nevertheless, we would like to highlight the potential for ultralow off-current enabled by the ultra-wide bandgap Ga₂O₃ channel material. The off-current density of the presented device is estimated to be only ~3fA/μm, limited by measurement instrument. Despite this limitation, the off-current density is still over five orders of magnitude lower than that of a modern low-power transistor with an off-current density of 1nA/μm. We believe that TFTs using the ultra-wide bandgap Ga₂O₃ offer a promising opportunity to achieve ultralow-power operation with scaled devices.

In the revised manuscript, we added the following sentence, “Since power consumption can be further reduced through scaling, the presented results demonstrate high potential

for ultra-low power consumption TFT technology. This is attributed to the low off-current characteristics originating from the ultra-wide bandgap nature.” for the NMOS inverter part and “These results reveal the potential for achieving low-power operation for oxide CMOS circuits using scaled devices.” for CMOS inverter part to highlight the potential of the Ga₂O₃ TFTs for low-power oxide circuits.

2. AC characterization: In the AC measurement section of our manuscript, our focus was solely on demonstrating the dynamic switch of the β -Ga₂O₃-based NMOS inverter, highlighting its potential for circuit applications. The device structure and dimensions for the AC measurement are consistent with those presented in **Figures 2 and 3**. The channel width (W) and length (L) are 300 μ m and 100 μ m, respectively, and the thickness of the gate oxide (SiO₂) is still 150 nm. However, due to the long channel device with a significant capacitance effect, discussing the AC characterization performance based on the presented results becomes less meaningful. The current speed of the AC characterization was limited by the device dimensions. Our ongoing and next important focus is on the circuit application using β -Ga₂O₃ TFTs. Therefore, we have decided to remove the discussion of the AC characterization performance in the revised manuscript. The sentence has been rewritten as follows. “The dynamic switching was also demonstrated to indicate the potential for circuit applications by further optimizing the dimensions of the devices.”

3. Inverter gain: Following the reviewer’s suggestion, we provide the average values of the gain of the inverter circuits. **Figures R13** (a) and (b) present additional data on the voltage transfer characteristic of the Ga₂O₃ TFT-based NMOS inverter and the *p*-SnO/*n*-Ga₂O₃ TFT-based CMOS inverter. **Figures R13** (c) and (d) display the average gain of the Ga₂O₃ TFT-based NMOS and *p*-SnO/*n*-Ga₂O₃ TFT-based CMOS inverter at different V_{DD}, respectively. The average voltage gain of the NMOS inverters was estimated as 16.9, 45.1, 64.0, 80.5, and 106.4 at V_{DD} from 5 to 25 V, respectively. Meanwhile, the average voltage gain of the CMOS inverters was estimated as 27.2, 40.3, 60.7, 90.7, and 124.9 at V_{DD} from 10 to 50 V, respectively.

We have added **Figure R13** to the supporting information (**Figure S33**). In the NMOS and CMOS inverter section of the revised manuscript, we revised the sentence as follows. “The average voltage gain of the NMOS inverters was 16.9, 45.1, 64.0, 80.5, and 106.4 at V_{DD} from 5 to 25 V, respectively, comparable to the previously reported oxide-NMOS inverters. (**Figure S33** for statistical results on the gain of the NMOS inverter, and **Table S4** for the NMOS inverter summary (Supporting Information))” and “The average voltage gain of the inverter was estimated as 38, 50, 79, 110, 149 at

V_{DD} from 10 to 50 V, respectively, which is nearly comparable to the previously reported values for oxide-based CMOS inverters. (**Figure S33** for statistical results on the gain of the CMOS inverter, and **Table S5** for the CMOS inverter summary (Supporting Information))”

Figure R13. (a) Voltage transfer characteristic of the β -Ga₂O₃ TFT-based NMOS inverter at $V_{DD} = 20$ V for 12 inverters. (b) Voltage transfer characteristic of the p -SnO/ n -Ga₂O₃ TFT-based CMOS inverter at $V_{DD} = 40$ V for 10 inverters. (c) Average gain of the Ga₂O₃ TFT-based NMOS at different V_{DD} . (d) Average gain of the p -SnO/ n -Ga₂O₃ TFT-based CMOS inverter at different V_{DD} .

Comment 10. *It is not immediately clear that the experimental setup shown in the supporting information is scalable to larger areas. The squeezing of a single droplet produces films that are only about 1 cm² in diameter. The authors need a compelling case for how this pressure assisted annealing could be scalable. Having a demonstration to show the variability across a substrate would also be important. The device statistics in this paper are relatively weak. The only histograms shown in Figure S4 have only 12 devices. Are these all from a single substrate? All of these details need to be clearly stated.*

(Response) Large-area β -Ga₂O₃ nanosheets, exceeding several centimeters in lateral dimensions, are achievable through the LMP method. The size of the nanosheet can also be expanded by using a greater quantity of liquid metal. Numerous studies in the literature report oxide nanosheets, including Ga₂O₃,^{12, 13} In₂O₃,¹ and ITO,¹⁴ with lateral dimensions exceeding several centimeters.

We understand the reviewer's concern regarding the scalability of the experimental setup, especially considering the limited size of the produced nanosheet. We believe that achieving large-scale oxide nanosheets prepared by PA-LMP is possible by designing a setup capable of applying wafer-scale and uniform pressure in the future. Our current focus involves the ongoing design of a printing tool to facilitate automatic and larger-scale printing processes.

Another critical concern is that the existing setup poses challenges in achieving high-yield and uniform device performance. This challenge may arise from non-uniform applied pressure, uncontrolled motion speed of the top substrate, and an experiment environment not controlled under ambient atmospheric conditions.

Regarding the demonstration of the variability of TFT performance across a substrate, we provide **Figure R14**, and **Figure R16** to show the TFT performance on a single substrate for the β -Ga₂O₃ TFT prepared by the P_P of 0.3 kgf/cm² and 1.3 kgf/cm² at the T_P of 150 °C, respectively. **Figure R14 (a)** and **Figure R16 (a)** display photographs of the TFT samples. We measured 4×4 arrays (16 devices) in the center of the sample with an area of ~10 × 10 mm². The average mobility of the devices prepared by the P_P of 0.3 kgf/cm² and 1.3 kgf/cm² were 1.84 cm²V⁻¹s⁻¹ and 5.05 cm²V⁻¹s⁻¹, respectively. However, the devices prepared by the higher P_P of 1.3 kgf/cm² show a large variation in device performance. Additionally, currently, the devices fabricated with a higher P_P of 1.3 kgf/cm² suffer from lower yield compared to those fabricated with the P_P of 0.3 kgf/cm². The optimization of the printing process, including the uniformity of pressed pressure and control of the experimental environment with controllable oxygen concentration and humidity, etc., would be important for further improvement in yield and uniformity in the future.

In order to enhance the device statistics in our manuscript, we also provide data on the batch-to-batch variation of β -Ga₂O₃ nanosheet TFTs. The nanosheets were prepared at T_P of 150 °C and P_P of 0.3 kgf/cm², as shown in **Figure R15**. For the Ga₂O₃ TFTs prepared with a higher P_P of 1.3 kgf/cm² at T_P of 150 °C, since we encountered a low yield of devices fabricated under the high process pressure condition, we present

selected devices on a substrate from a different batch to demonstrate the reproducibility of β -Ga₂O₃TFT device performance (**Figure R17**).

We have added **Figure R14-16** to the supporting information. Also, in the revised manuscript, we revised the following sentence, “The device-to-device statistical analysis was also performed using 10 working devices on a single substrate, and the results are summarized in **Figure S7** (Supporting Information). The average values of μ_{sat} of $5.02 \pm 4.05 \text{ cm}^2 \text{ V}^{-1} \text{ s}^{-1}$, μ_{lin} of $4.21 \pm 5.87 \text{ cm}^2 \text{ V}^{-1} \text{ s}^{-1}$, s -values of $0.19 \pm 0.12 \text{ V} \cdot \text{dec}^{-1}$, V_{th} of $3.61 \pm 2.99 \text{ V}$, and $\log(I_{on}/I_{off})$ of 8.09 ± 1.54 were obtained. (**Figure S7-9** (Supporting Information) provides a detailed discussion about the device uniformity)”

Figure R14. (a) Photographs of β -Ga₂O₃ TFT sample. (b) Transfer characteristics for the Ga₂O₃ nanosheet TFTs for 15 devices on a single substrate. (The β -Ga₂O₃ nanosheet channel was prepared at the T_p of 150 °C and the P_p of 0.3 kgf/cm².) Histograms of the device performances obtained from device-to-device statistical analysis for (c) μ_{sat} , (d) s -value, (e) V_{th} , and (f) on/off-current ratio of the atomically thin β -Ga₂O₃ TFT.

Figure R15. Batch-to-batch variation of $\beta\text{-Ga}_2\text{O}_3$ TFTs: Ga_2O_3 nanosheets were prepared at the T_p of $150\text{ }^\circ\text{C}$ and the P_p of 0.3 kgf/cm^2 . (The working devices in Batch 1 were 15, in Batch 2 were 13, and in Batch 3 were 14.) (a) μ_{sat} , (b) s -value, (c) V_{th} , and (d) on/off-current ratio, respectively.

Figure R16. (a) Photographs of the β -Ga₂O₃ TFT device. (b) Transfer characteristics for the Ga₂O₃ nanosheet TFTs for 10 devices on a single substrate. (The Ga₂O₃ nanosheet was prepared at the T_p of 150 °C and the P_p of 1.3 kgf/cm²) Histograms of the device performances obtained from the device-to-device statistical analysis for (c) μ_{sat} , (d) s -value, (e) V_{th} , and (f) on/off-current ratio, respectively.

Figure R17. Transfer characteristics of selected 5 working β -Ga₂O₃ TFTs on a single chip substrate. The Ga₂O₃ nanosheets were prepared at the T_p of 150 °C and the P_p of 1.3 kgf/cm². (a) Batch 1, (b) Batch 2, (c) Batch 3.

Comment11. *The other bit of context that is needed for understanding the crystallization of the beta-Ga₂O₃ phase here is a comparison to the other reports of beta-Ga₂O₃ formed by liquid metal printing (e.g. 10.1038/s41699-021-00219-y)*

(Response). The reference (doi.org 10.1038/s41699-021-00219-y)⁴, demonstrating the direct growth of crystalline β -Ga₂O₃ by LMP provides us deeper understanding of growing crystallized Ga₂O₃ by PA-LMP. Direct synthesis of β -Ga₂O₃ by LMP has also been reported in doi.org/10.1016/j.matchemphys.2021.125652). Another work (doi.org/10.1016/j.matchemphys.2021.125652) also synthesized β -Ga₂O₃ by liquid metal printing using liquid Galinstan, but the detailed mechanism is still unclear. However, this report also clearly indicates that the LMP method can directly facilitate the growth of crystalline β -Ga₂O₃.¹⁵

Furthermore, several literatures have been demonstrated in a variety of high-quality oxide materials such as β -PbO¹⁶, α -Bi₂O₃¹⁷, and β -TeO₂.¹⁸ Based on the literature suggested in comment 2 by Reviewer (10.1038/s41699-022-00294-9), we consider that the nature of oxidation of the liquid metal present distinct thermodynamic barrier, allowing the process to bypass the formation of intermediate phases. For the formation of β -Ga₂O₃, the traditional process involves the formation of GaO hydrate and gel forms and their transformation into the β -phase during subsequent processing. In contrast, the LMP takes direct pathways for growing oxide materials, potentially making it easier to produce high-quality oxide materials at lower process temperatures compared to other material growth approaches. Additionally, external stimuli, such as heat and stress, could facilitate the crystallization of the β -Ga₂O₃ phase.

The revised section in the manuscript: “We believe that the nature of oxidation of the liquid metal has a different thermodynamic barrier and path to grow oxide materials, potentially making it easier to grow high-quality oxide materials at lower process temperatures compared to other material growth approaches. Crystalline β -Ga₂O₃ has been successfully synthesized using liquid metal printing, achieved by scraping off the parent metal layer under stress at a process temperature of 200°C. Given that the thermodynamic barrier and pathway may lower when employing liquid metal oxidation, the crystallization of the Ga₂O₃ phase could potentially be induced by various factors, such as heat, pressure, and stress, to lower the energy barriers for crystallization.”

Comment 12. *Does the printing pressure influence the thickness or uniformity of the transferred films? Is the morphology similar between the two? Microscope images*

showing the nanosheet morphology more in detail compared with what is currently in Figure S2 would be helpful.

(Response) In response to reviewer's comment, we conducted atomic force microscopy (AFM) analysis on β -Ga₂O₃ nanosheets on SiO₂/Si substrate samples under different conditions: (a) the T_p of 80°C with different P_p conditions (i.e, no applied pressure (0), 0.3, and 1.3 kgf/cm²). (**Figure R18**) (b) the T_p of 150°C with different printing process pressures. (See **Figure R19**) No significant difference in the thickness was observed under low applied pressure conditions. However, we observed a slight decrease in thickness with increasing pressure. Regarding surface roughness, we observed that the applied pressure resulted in a flatter surface, with the surface roughness decreasing as the applied pressure increased.

Figure R20 also presents the photographs and optical microscope images of large-scale (cm-sized) β -Ga₂O₃ nanosheets fabricated by PA-LMP at T_p of 150 °C under different pressure conditions. (i.e., no pressure, 0.3, 1.3, and 2.1 kgf/cm²) We observed that all of these samples exhibited large-scale Ga₂O₃ nanosheets with dimensions of approximately 1×1 cm². Moreover, we can confirm that the entire laterally large Ga₂O₃ layer is continuous and free of significant holes and cracks.

Following the reviewer's suggestion, therefore, we have added **Figure R18, R19, R20** and provided additional descriptions in the main text in the revised main text and supporting information. (**Figure S4, S30, and S31**) The revised section in the manuscript. "The optical microscope images of these samples with different process conditions, which we investigated for their electrical properties, are also shown in **Figure S4**. (Supporting Information) Furthermore, the AFM image reveals the effect of the applied pressure on the surface morphology, as discussed in the **Supporting Information**. (**Figure S30, S31**, Supporting Information)."

Figure R18. AFM image and the cross-sectional step-height profile for the $\beta\text{-Ga}_2\text{O}_3$ nanosheet were prepared by PA-LMP methods at the T_p of 80 °C with different printing process pressure conditions (a) w/o (b) 0.3 kgf/cm², and (c) 1.3 kgf/cm².

Figure R19. AFM image and the cross-sectional step-height profile for the $\beta\text{-Ga}_2\text{O}_3$ nanosheet were prepared by PA-LMP methods at the T_p of 150 °C with different printing process pressure conditions (a) w/o (b) 1.3 kgf/cm², and (c) 2.1 kgf/cm².

Figure R20. Photographs and optical microscope images for a large-scale (cm-sized) β - Ga_2O_3 nanosheet fabricated by PA-LMP at the T_p of 150 °C under different pressure conditions: (a) without pressure. (b) 0.3 kgf/cm², (c) 1.3 kgf/cm², (d) 2.1 kgf/cm².

Comment 13. *The argument regarding the activation energy for crystallization on pg. 10 and 11 is interesting – it would have been helpful to have a sense for how the annealing time influences the crystallinity. Does a long term of applied pressure increase the crystallinity? Why was 3 minutes chosen?*

(Response) We have investigated the impact of the process duration under applied pressure for 1, 3, and 5 minutes. (T_p of 150 °C and P_p of 0.3 kgf/cm²) We observed that the device mobility remained nearly constant, as shown in **Figure R21** (selected working device), and there was no significant difference. When we further extended the processing time beyond 10 minutes, however, the channel exhibited insulating behavior and rendered the TFT inoperative. Based on these results, we suspected that when the processing time is too long, the carrier concentration of the Ga_2O_3 becomes too low due to a decrease in oxygen vacancies. Furthermore, our preliminary results indicate that

prolonged processing time does not yield further improvement in crystallinity, thus failing to enhance TFT device performance. Further investigation is required for future studies.

Following the reviewer's suggestion, we have provided additional descriptions in the experimental section of the main text in the revised manuscript. Additionally, we have included **Figure R21** in the supporting information.

The revised section in the manuscript: "When the printing process time was too long (>10 min), the channel exhibited insulating behavior, rendering the TFTs inoperative. (**Figure S36**, Supporting Information)"

Figure R21. Variation of the saturation mobility by different printing process time. A long printing process time (>10 min) cannot produce operative Ga₂O₃ nanosheet TFTs.

Other comments

1) *There is no color bar to give an idea about the height map scale for parts a and b of Figure 2. The line scan itself is very small.*

(Response) We have improved Figure 2 (a), (b).

2) *Would be good to have small English edits to details such as the figure captions*

(Response) We have improved the figure captions.

Reviewer #2 (Remarks to the Author):

The manuscript 'Low-temperature crystallization in nanoscale confinement enabled by pressure-assisted liquid-metal printing for β -Ga₂O₃ thin-film transistors' follows on from a significant published body of work in the emerging area of liquid metal printed 2D materials. The deposition of gallium oxide nanosheets from liquid metals is well known and ~30+ papers have appeared on this topic, either focusing on the direct use of the grown Ga₂O₃ or using it as a building block to make other 2D materials. Furthermore, many other 2D materials have been developed using liquid metal printing with SnO, SnO₂, ITO, and In₂O₃ showing impressive electronic device performances sometimes exceeding what has been reported in this manuscript. As such, there are some concerns reading the impact and novelty, and whether the presented work is suitable for an outlet such as Nature Communications. This is ultimately a decision for the editors. (highligh novelty in Ga₂O₃ TFT)

Aside from the synthesis of Ga₂O₃, the authors report the application of mild pressure and heat that is claimed to facilitate crystallisation, leading to a variety of gallium-based oxides including beta Ga₂O₃, amorphous Ga₂O₃ and Ga₂O. Overall this aspect of the work is not convincing. The applied forces are too low to be consequential. 0.3 kgf/cm² is roughly similar to what can be applied by hand. Pressures of ~2 kgf/cm² are only slightly higher. This is almost certainly not the reason for crystallization, since printing by hand is widely used in the literature using similar pressures while leading to amorphous films. Assuming that the materials do indeed crystallize, it is more likely that the reason triggering crystallization is a different one. Maybe the change in surface chemistry of the substrate after plasma cleaning? Or maybe another minor change to the procedure. Maybe there are minor impurities in the used materials that cause the effect? Irrespective of this, the applied energy is very unlikely to cause the crystallisation.

The authors should conduct proper calculations determining the applied energy in units of J/mol and discuss if this may be a sufficient activation energy. Furthermore, the evidence provide for the apparent crystallisation of the material is not conclusive. The authors only provide a TEM image. This is not sufficient. First of all, how did the authors prepare the TEM sample? The TEM membrane is usually soft and suspended inside a thicker copper mesh. Applying pressure from above is not likely to actually be exerted onto the Ga₂O₃ sheet. In order to convincingly show that the material has been crystallized, the authors should provide data such as:

1) Raman mapping, 2) 4D STEM (or nanobeam diffraction), 3) Atomic resolution AFM or STM at various positions on a single sheet. This should be conducted for a range of samples, showing that the absence of pressure leads to amorphous films, while the presence leads to crystallization. The supposed synthesis of Ga₂O (or Ga(I) oxide) has not been properly confirmed at all. This material should be thoroughly characterised since it would be a significant achievement. The authors proposed mechanism hinges on several assumptions. One of the assumptions is that the volume of beta gallium oxide is smaller than that of the amorphous film. This is not a forgone conclusion. The Authors need to proof this. Furthermore, in order to be convincing, the authors would need to conduct DFT calculations showing that the proposed reaction is indeed feasible and can indeed be triggered by such a minimal activation energy provided by the pressure. If this was indeed the case, then crystallisation of the amorphous Ga₂O₃ phase would be easily triggered by many other stimuli including heat, which is evidently not the case (literature reports have shown heating of amorphous Ga₂O₃ to several hundred C without observing crystallisation).

Minor notes: The equation in line 153 seems to be incorrect. The free energy change (ΔG) is dependent on the change of the enthalpy (ΔE) and the change of then entropy (ΔS). The deltas have been omitted on the right-hand side of the equation. The relevance of Figure 1 a is not clear. It might be more useful to utilize this space for something more relevant.

(Response) Thank you for the reviewer’s thoughtful comments and constructive suggestions. We appreciate the opportunity to address their concerns and provide point-to-point responses along with additional experimental data to support our findings further.

Comment 1. *The deposition of gallium oxide nanosheets from liquid metals is well known and ~30+ papers have appeared on this topic, either focusing on the direct use of the grown Ga₂O₃ or using it as a building block to make other 2D materials. Furthermore, many other 2D materials have been developed using liquid metal printing with SnO, SnO₂, ITO, and In₂O₃ showing impressive electronic device performances sometimes exceeding what has been reported in this manuscript. As such, there are some concerns reading the impact and novelty, and whether the presented work is suitable for an outlet such as Nature Communications. This is ultimately a decision for the editors. (highligh novelty in Ga₂O₃ TFT)*

(Response) We acknowledge that there are several reports on gallium oxide nanosheets produced by the LMP route, but most of them only reported insulative amorphous Ga₂O₃. We also know the literature (DOI: 10.1038/s41699-021-00219-y) reporting crystalline β -Ga₂O₃ by the LMP technique at a process temperature of 200°C.⁴ However, they have not successfully fabricated thin-film electronic devices using β -Ga₂O₃ channel, which is crucial for the development of high-performance electronics due to superior material properties, such as a largest wide-band gap nature (> 4.5 eV) in the oxide semiconductor potential high mobility and the excellent material stability., enabling various applications using low-temperature processed Ga₂O₃ thin-film electronics.

Therefore, the intense effort on the development of electronic devices using β -Ga₂O₃ is ongoing. However, the current research on β -Ga₂O₃ device application is mainly based on metal-oxide-semiconductor field-effect transistor (MOSFET) structure using a single-crystal wafer for applications in power electronics, due to the challenging of film growth of high-quality β -Ga₂O₃ thin film (**Figure R22**). In general, conventional PVD deposition methods require high growth temperatures exceeding 600 °C to grow semiconducting Ga₂O₃, which are not acceptable for many TFT device applications. Therefore, the development of ultra-wide bandgap β -Ga₂O₃ by a low-temperature process is highly demanded for next-generation electronics.

Figure R22. Motivation for the development of the low-temperature processed β -Ga₂O₃ thin-film electronics.

In our response to the reviewer’s comment, *“Furthermore, many other 2D materials have been developed using liquid metal printing with SnO, SnO₂, ITO, and In₂O₃ showing impressive electronic device performances sometimes exceeding what has been reported in this manuscript.”* We believe that this comment is not fair and not entirely accurate. First, SnO is a *p*-type oxide material, and it serves a completely different device application. It is true that many reports on ITO, and In₂O₃ TFTs fabricated by the LMP route demonstrated better device performances than conventional PVD-produced devices. However, conventional PVD-produced devices can also yield satisfactory ITO and In₂O₃-TFTs. In contrast, the β -Ga₂O₃ TFT development is largely behind the others, and the TFT device performance is very poor. This work presents significant advancements in β -Ga₂O₃ TFT development and a high-potential of β -Ga₂O₃ for electronic device applications. **Table S1** in the supporting information summarizes the device performance for previously reported Ga₂O₃ TFTs for comparison.

Following the reviewer’s suggestion to highlight the novelty, motivation, and advancement of developing the low-temperature processed Ga₂O₃ thin-film electronics, we revised the main text and added **Figure R21** in **Figure 1**.

The revised section in the manuscript: “Current research on β -Ga₂O₃ transistors primarily focuses on metal-oxide-semiconductor field-effect transistor (MOSFET) structures using bulk single-crystal wafers for power devices. However, developing β -Ga₂O₃ TFTs remains challenging due to the high-temperature process required for β -Ga₂O₃ film growth in conventional physical vapor deposition (PVD) methods. Nevertheless, low-temperature processes for Ga₂O₃ TFTs show promise for various practical applications, such as low-power switching devices and solar-blind deep-ultraviolet photodetectors, originating from their ultra-wide bandgap nature. (Figure 1(a))”

“Importantly, the presented β -Ga₂O₃ nanosheet TFTs exhibit higher mobility and lower D_{it} with a much lower process temperature compared to reported Ga₂O₃ TFTs fabricated by traditional thin-film processing.”

Comment 2 *Aside from the synthesis of Ga₂O₃, the authors report the application of mild pressure and heat that is claimed to facilitate crystallisation, leading to a variety of gallium-based oxides including beta Ga₂O₃, amorphous Ga₂O₃ and Ga₂O. Overall this aspect of the work is not convincing. The applied forces are too low to be consequential. 0.3 kgf/cm² is roughly similar to what can be applied by hand. Pressures of ~2 kgf/cm² are only slightly higher. This is almost certainly not the reason for crystallization, since printing by hand is widely used in the literature using similar pressures while leading to amorphous films.*

(Response) To address the reviewer’s concerns, we have provided additional TEM analysis for the Ga₂O₃ nanosheet on a TEM grid, as well as cross-section TEM analysis of the nanosheet directly on the substrate (device sample), and the GIXRD analysis to confirm the crystalline β -Ga₂O₃ phase. All the data and discussion have been made in in our response to **Reviewer 1 (comment 1), and Reviewer 2 (comment 4 and 5)**. Please refer to the section for the detailed information.

-Regarding the comment, “The TEM membrane is usually soft and suspended inside a thicker copper mesh. Applying pressure from above is not likely to actually be exerted onto the Ga₂O₃ sheet.”

(Response) We agree with this comment, as it highlights a critical issue in many current papers discussing liquid metal-printed Ga₂O₃. Most of the literature on Ga₂O₃ nanosheets grown by LMP utilizes TEM analysis of the nanosheet on a TEM grid. As **Reviewer 2** mentioned, applying the same pressure/force for the TEM sample as with

the nanosheet on other substrates is not feasible. Thus, the TEM materials characterization in these reports may not accurately reflect the real situation for the materials fabricated on the substrate and could be considered less reliable. Even if they have synthesized crystalline Ga₂O₃ materials, they may not be able to detect them using the nanosheet on a TEM grid sample.

Considering that the nanosheet samples placed on TEM grids may differ from those fabricated on the substrate, it may be less meaningful to use these literature references to counter our findings. The TEM results from these references might not be entirely convincing or representative of the samples grown on substrates. Furthermore, even if these reports cannot verify the crystallinity based on XRD, it does not necessarily indicate that their Ga₂O₃ is amorphous since the Ga₂O₃ nanosheets are too thin.

-Most importantly, the reviewer's comment, "*printing by hand is widely used in the literature using similar pressures while leading to amorphous films.*" is not completely correct. There are many reports that have used touch printing or rolling printing without applying pressure. (we discuss in the **Figure S3** in the supporting information) Moreover, the literature (10.1038/s41699-021-00219-y) have already demonstrated LMP for the growth of crystalline β -Ga₂O₃.⁴ Another work (doi.org/10.1016/j.matchemphys.2021.125652) also synthesized β -Ga₂O₃ by liquid metal printing using liquid Galinstan.¹⁵

In our revised manuscript, we believe that comprehensive TEM analysis of the nanosheet on TEM grid samples, cross-section TEM analysis of the nanosheet directly on the substrate, and XRD analysis provide solid and convincing data to demonstrate that crystallized β -Ga₂O₃ can be grown using the proposed liquid metal printing approach.

Comment 3. *Assuming that the materials do indeed crystallize, it is more likely that the reason triggering crystallization is a different one. Maybe the change in surface chemistry of the substrate after plasma cleaning? Or maybe another minor change to the procedure. Maybe there are minor impurities in the used materials that cause the effect? Irrespective of this, the applied energy is very unlikely to cause the crystallisation. The authors should conduct proper calculations determining the applied energy in units of J/mol and discuss if this may be a sufficient activation energy.*

(Response) Thank you for reviewer’s insightful comments and concerns. We believe this discussion is off from our topic in this paper. Nevertheless, we recognized that the theoretical calculation approach to discuss the activation energy for crystallization is important. Therefore, collaboration with material and process theorists may be necessary for this purpose in the future.

To address the reviewer’s concern, we add the following discussion in the revised manuscript. The revised section in the manuscript, “The liquid metal printing approach can form crystalline and highly conductive oxide films upon deposition, avoiding insulating intermediate phases and eliminating the thermodynamic barriers posed by precursor decomposition, which is typical in the traditional approach to growing β -Ga₂O₃. The traditional processes involve the formation of GaO hydrate and gel forms, which then transform into the β -phase during subsequent processing. **(Figure 1(b))** We believe the oxidation process of the liquid metal presents different thermodynamic barriers and pathways for growing oxide materials, potentially making it easier to produce high-quality oxide materials at lower process temperatures.”

“We believe that the nature of oxidation of the liquid metal has a different thermodynamic barrier and path to grow oxide materials, potentially making it easier to grow high-quality oxide materials at lower process temperatures compared to other material growth approaches. Crystalline β -Ga₂O₃ has been successfully synthesized using liquid metal printing, achieved by scraping off the parent metal layer under stress at a process temperature of 200°C. Given that the thermodynamic barrier and pathway may lower when employing liquid metal oxidation, the crystallization of the Ga₂O₃ phase could potentially be induced by various factors, such as heat, pressure, and stress, to lower the energy barriers for crystallization.”

Comment 4, *Furthermore, the evidence provide for the apparent crystallisation of the material is not conclusive. The authors only provide a TEM image. This is not sufficient. First of all, how did the authors prepare the TEM sample? The TEM membrane is usually soft and suspended inside a thicker copper mesh. Applying pressure from above is not likely to actually be exerted onto the Ga₂O₃ sheet. In order to convincingly show that the material has been crystallized, the authors should provide data such as: 1) Raman mapping, 2) 4D STEM (or nanobeam diffraction) , 3) Atomic resolution AFM or STM at various positions on a single sheet. This should be conducted for a range of samples, showing that the absence of pressure leads to amorphous films, while the presence leads to crystallization.*

(Response) 1. TEM sample preparation. We provide detailed procedures and tips to prepare the TEM sample and perform the TEM analysis. We have also included this information in the revised supporting information.

2. TEM sample preparation procedure. The TEM grid was placed on top of the glass slide and pre-heated to process temperature (80–200 °C), which is above the melting point of the liquid Ga metal. Note that we did not perform O₂ plasma treatment for the TEM sample, as we did for the Ga₂O₃ nanosheet on the SiO₂/Si substrate.

To fabricate the nanosheets, a liquid droplet of Ga metal (with a size of <1 mm) was pipetted onto a TEM grid, and the TEM grid/glass slide with the liquid metal droplet was kept heated at 50°C to prevent the liquid Ga from solidifying. During the printing process, we used SiO₂/Si as the top substrate (pre-heated to process temperature (80–200 °C)), pressing it onto the center of the droplet to spread the liquid alloy homogeneously between the TEM grid and SiO₂/Si substrates. The TEM grid and SiO₂/Si substrates were kept at process temperature (80–200 °C) under uniaxial vertical pressure for gallium oxide nanosheet growth for 3 minutes. After the squeezing step with uniaxial vertical pressure, the top SiO₂/Si substrate and TEM grid were carefully separated. We then used a soft wiping tool (cotton bud) to remove Ga liquid inclusions directly by gentle rubbing. It is important to note that we did not immerse the sample in ethanol, as we did for the Ga₂O₃ nanosheet on the SiO₂/Si substrate.

Following the printing process, we observed that most carbon film was broken (**Figure R23**). During TEM analysis, we looked for nanosheets suspended and connected/supported by the bar of the TEM grid for detailed analysis (the red rectangle region in **Figure R23**).

We have provided the detailed TEM sample preparation procedure in the method section of the main manuscript and supporting information in the revised manuscript (**Figure S2**).

Figure R23. Optical microscopy image of Ga_2O_3 nanosheet on the TEM grid.

3. Material characterization. We have conducted further TEM analysis of the Ga_2O_3 nanosheet prepared by the PA-LMP and provided the data and discussion in our response for **comment 1 in Reviewer 1.**

4. Regarding the materials characterization methods suggested by Reviewer 2, including 1) Raman mapping, 2) 4D STEM (or nanobeam diffraction), and 3) Atomic resolution AFM or STM. We consider that Raman mapping and atomic resolution 4D STEM, atomic resolution AFM, and STM, are not typically used as direct methods to prove the crystallinity of materials, and they may not be necessary for this purpose. We believe that XRD and TEM are more commonly used for this purpose. Furthermore, the atomic resolution STM is not a common tool in most research labs. 4D STEM is a relatively new technology that is still under development and may not be widely available in most research institutions.

Comment 5, *The supposed synthesis of Ga_2O (or Ga(I) oxide) has not been properly confirmed at all. This material should be thoroughly characterized since it would be a significant achievement.*

(Response) We also appreciate the reviewer's interest in the Ga(I) oxide materials. Our material characterization based on TEM analysis (**Figure R7**) showed that Ga_xO is amorphous. Importantly, we would like to highlight that our focus is the demonstration of the low-temperature processing of wide-bandgap $\beta\text{-Ga}_2\text{O}_3$ for thin-film transistor applications. We mentioned the preliminary results of *p*-type Ga(I) oxide to generate

interest in the oxide thin-film transistor community and encourage exploration of these new materials. Additionally, further research investigating *p*-type Ga(I) oxide for *p*-type oxide transistors is of interest to us and is part of our future work. We hope the reviewer will focus on our main claim and achievement regarding the low-temperature processed wide-bandgap β -Ga₂O₃ for thin-film transistors rather than the extended future work.

Comment 6, *The authors proposed mechanism hinges on several assumptions. One of the assumptions is that the volume of beta gallium oxide is smaller than that of the amorphous film. This is not a forgone conclusion. The Authors need to proof this. Furthermore, in order to be convincing, the authors would need to conduct DFT calculations showing that the proposed reaction is indeed feasible and can indeed be triggered by such a minimal activation energy provided by the pressure. If this was indeed the case, then crystallisation of the amorphous Ga₂O₃ phase would be easily triggered by many other stimuli including heat, which is evidently not the case (literature reports have shown heating of amorphous Ga₂O₃ to several hundred C without observing crystallisation).*

(Response) Based on the literature (doi:10.1038/am.2017.20 and doi.org/10.1063/5.0159529), the amorphous GaO_x film density was determined to be approximately 5.2 – 5.4 g·cm⁻³, which is lower than that of β -Ga₂O₃ (density ~5.95 g·cm⁻³)^{11, 19}. Following the reviewer's suggestion, we have added the comment and reference to the main manuscript.

The revised section in the manuscript: “The amorphous GaO_x film density was determined to be approximately 5.2–5.4 g·cm⁻³, which is lower than that of β -Ga₂O₃ (density ~5.95 g·cm⁻³).”

We also appreciate the reviewer’s constructive insights, which have helped us in explaining the mechanism for the low-temperature growth of crystalline Ga₂O₃. Our response has been made already. Please refer to **Reviewer 1 (comment 2 and 11)**.

- If this was indeed the case, then crystallisation of the amorphous Ga₂O₃ phase would be easily triggered by many other stimuli including heat, which is evidently not the case.

(Response) This is an open question, and it appears to be the case based on the literature mentioned above.⁴ (10.1038/s41699-021-00219-y)

-literature reports have shown heating of amorphous Ga₂O₃ to several hundred C without observing crystallization.

(Response) The oxidation process of liquid metal differs from the phase transformation from an amorphous GaO_x film to crystalline Ga₂O₃. Additionally, it is well-known that the crystallization temperature strongly depends on the deposition/preparation method of the amorphous structure of GaO_x. The reviewer's comment cannot deny the low-temperature crystallization of Ga₂O₃. Given this, it may not be appropriate to use this case as an argument against our results.

Comment 7. *Minor notes: The equation in line 153 seems to be incorrect. The free energy change (Delta G) is dependent on the change of the enthalpy (delta E) and the change of then entropy (delta S). The deltas have been omitted on the right-hand side of the equation.*

(Response) Thank you for your careful review. We have corrected this part.

Comment 8. *The relevance of Figure 1 a is not clear. It might be more useful to utilize this space for something more relevant.*

(Response) Following the suggestion from the reviewer, we have added **Figure R22** to **Figure 1(a)** to highlight the motivation of Ga₂O₃ thin-film transistors. Additionally, we have included the following sentence to describe the advantages of liquid metal printing for the growth of low-temperature oxide materials. Here, we refer to **Figure 1(b)** (original **Figure (a)**) to state that the liquid metal printing approach can form crystalline and highly conductive oxide films upon deposition, avoiding insulating intermediate phases and eliminating the thermodynamic barriers posed by precursor decomposition, which are typical in the traditional approach to growing β -Ga₂O₃. The traditional processes involve the formation of GaO hydrate and gel forms, which then transform into the β -phase during subsequent processing.

The revised section in the manuscript: “The liquid metal printing approach can form crystalline and highly conductive oxide films upon deposition, avoiding insulating intermediate phases and eliminating the thermodynamic barriers posed by precursor decomposition, which is typical in the traditional approach to growing β -Ga₂O₃. The traditional processes involve the formation of GaO hydrate and gel forms, which then transform into the β -phase during subsequent processing. **(Figure 1(b))** We believe the oxidation process of the liquid metal presents different thermodynamic barriers and

pathways for growing oxide materials, potentially making it easier to produce high-quality oxide materials at lower process temperatures.”

Reviewer #3 (Remarks to the Author):

In this work, the author introduces a pressure-assisted strategy on the reported liquid metal squeezing method to effectively induce the direct transition of 2D Ga₂O₃ from amorphous to polycrystal state at a low temperature (< 200 °C). Moreover, the pressure-assisted strategy simultaneously enables the change of electrical properties from insulator (amorphous 2D Ga₂O₃) to n-type semiconductor (polycrystalline 2D β-Ga₂O₃). The fabricated 2D β-Ga₂O₃ TFTs exhibit superior switching properties with high saturation mobility and large on/off-current ratio (~10⁹). The finding is interesting and useful for the design of high-performance oxide TFTs. However, we only have one concern that such a low pressure of 0.3 kgf/cm² can really promote the crystallization of amorphous Ga₂O₃ at a low temperature of 80 °C. Actually, TEM analysis is not adequate to support the formation of β-Ga₂O₃ and more crystal structure characterizations should be tested. In my opinion, this manuscript could not be recommended to publish before the following issues are properly addressed.

(Response) Thank you for your thoughtful comments and constructive suggestions. We have revised our manuscript and provided additional data to address your detailed comments.

Comment1. *It is challenging to exfoliate and transfer 2D Ga₂O₃ film from substrate to TEM grid. A detailed description of the preparation/transfer process of TEM sample is easy for readers to refer and repeat. The authors should add corresponding detailed description in the experimental section.*

(Response) We agree with the reviewer's concerns regarding TFM sample preparation. We have addressed these concerns in our response, which is provided in **comment 4 in Reviewer 2**. Please refer to that for detailed information.

Comment2. *Figure 2j shows a distinct crystalline feature. However, it is only a small region with 10 nm *10 nm. Whether 2D Ga₂O₃ amorphous film is completely transformed into polycrystal crystal should be further confirmed. SAED tests are suggested. Is it possible to exist a local region of amorphous Ga₂O₃ in the large-scale film? If yes, how does it influence the electrical performance of 2D Ga₂O₃ film? In addition, the observed crystallinity of Ga₂O₃ may come from the irradiation of electron beam which also produces heat to induce the phase transition. To fully examine the crystallization of 2D Ga₂O₃, it is necessary to conduct the HRTEM (SAED) analysis of large-area samples in short time.*

(Response) Following the reviewer's suggestions, we conducted a wide area of cross-sectional TEM characterization with SAED analysis for the Ga₂O₃ nanosheets grown at the T_p of 150 °C and the P_p 1.3 kgf/cm². We have addressed these concerns in our response, which was provided in **comment 1 in Reviewer 1**. Please refer to that for detailed information.

We also appreciate the insightful comment regarding the possibility of the existence of a local region of amorphous GaO_x in the large-scale film. From our TEM analysis, we have confirmed the presence of crystalline Ga₂O₃ over a large area, as shown in **Figure R4** and **R5**. Nevertheless, we cannot completely omit the possibility of the amorphous phase in the local regions. If amorphous GaO_x exists in the channel, we consider that mobility and uniformity would be degraded.

Regarding the crystallization induced by electron beam irradiation, we consider it unlikely to occur in the presented case. It is generally known that electron beam irradiation damages the material, leading to the degradation of crystallinity and causing the crystalline film to become amorphous. Therefore, the electron beam would not trigger a phase transition, converting the amorphous state into crystalline Ga₂O₃.

Comment 3. *In Figure 2i-k, the quality of TEM images is poor. It is difficult to confirm the formation of β-Ga₂O₃ from these results. A detailed and perfect TEM test should be required. Additionally, a series of XRD, Raman, XPS and PL tests are also required. It is significantly important to completely verify the formation of β-Ga₂O₃ crystal because it is the core finding of this work. Reliable experimental evidences should be provided.*

(Response) Following the reviewer's comment, we have provided a comprehensive TEM analysis (Ga₂O₃ nanosheet on TEM grid for the Ga₂O₃ prepared under different process parameters and cross-sectional TEM analysis of the Ga₂O₃ nanosheet on SiO₂/Si substrate) in response to **Reviewer 1 (comment 1)**. The GIXRD characterization was also performed and provided in the response to **Reviewer 1 (comment 1)**.

Moreover, we also performed X-ray photoemission spectroscopy (XPS) analysis. **Figure R24** presents the core spectra of the Ga 3d and O 1s for the Ga₂O₃ nanosheet grown by PA-LMP with the T_p of 150 °C and the P_p of 1.3 kgf/cm². For the XPS sample,

we prepared the Ga₂O₃ nanosheet on a Si substrate to avoid the charge-up effect, as well as the O signal from the substrate. The Ga 3d core level exhibited a distinct peak at the position of 19.7 eV, corresponding to Ga³⁺ oxidation state. The characteristic peaks observed at 531.5 eV are attributed to the O 1s spectra of the Ga(III)-O bond in Ga₂O₃.

Figure R24. XPS spectra of (a) Ga 3d and (b) O 1s core levels for the Ga₂O₃ nanosheet. (Ga₂O₃ nanosheet synthesized by the pressure-assisted liquid-metal route with the T_p of 150 °C and the P_p of 1.3 kgf/cm².)

We believe that the provided data, including comprehensive TEM analysis of the nanosheet on TEM grid samples, cross-section TEM analysis of the nanosheet directly on the substrate, and XRD analysis, address the reviewer's concern regarding the formation of crystalline β -Ga₂O₃ by the proposed PA-LMP approach.

Furthermore, we acknowledge that Raman and PL could provide additional insights into the material structure. However, we consider them unnecessary for our current purpose of studying the electrical properties of low-temperature processed β -Ga₂O₃ thin-film transistors.

Comment 4. *In Figure S8a, the presented STEM image is not clear. In Figure S8d, the EDX spectrum has several un-indexed peaks. Which elements do they belong and what effect they will have to affect the electrical properties in such thin layer?*

(Response) Following the reviewer's suggestion, we assigned all peaks in the EDX spectrum. **(Figure R25)** Besides the peaks for Ga and O, we observed peaks index to C, Cu, Si, K, and Ca. The C and Cu peaks predominantly originate from the TEM grid. On the other hand, the Si may result from the materials preparation process because we

used SiO₂/Si as the top substrate for the printing process. Additionally, the Si signal might be due to cross-contamination from the tweezers. The origin of K and Ca are not clear, but we believe they are attributed to airborne contaminants because our process is conducted in ambient air.

Confirming the influence of elements such as Si, K, and C on doping or other effects that may impact electrical properties proves challenging at the present stage, we believe that future work is necessary to clarify the impact of these elements on the electrical properties of Ga₂O₃ materials through theoretical and experimental approaches. Given the characteristics of the electronic structure of ionic oxide materials, the impact of such contamination may be relatively minimal.

We would like to emphasize that all experiments were conducted in the same laboratory environment, employing identical procedures for preparing Ga₂O₃ material samples. The only variable parameter was printing pressure in LMP method. Consequently, we conclude that the differences in electrical properties under different process conditions are not a result of the elements (Si, K, Ca) originating from the fabrication process or the environment.

We have added the above discussion to the EDX data section in the supporting information in the revised manuscript. (**Figure S18**)

Figure R25. EDX spectrum for the Ga₂O₃ nanosheet. The inset shows the atomic ratio of the Ga₂O₃ nanosheet.

Comment 5. The author mentioned that “when the device was fabricated at 30 °C, the channel showed highly conducting behavior with an electrical conductivity of 28.3 S·m-

1, but the device exhibited negligible field-effect current modulation". What's the reason? Does the film have the high content of residual metallic Ga? XPS tests are suggested.

(Response) Our observations suggest that the low-process temperature process at 30 °C with 0.3 kgf/cm² may not be sufficient to form Ga₂O₃ phases. Moreover, a low-temperature process causes the existence of residual Ga metal impurity and the formation of high-density oxygen vacancy, resulting in highly conducting behavior. The presence of residual Ga metal was confirmed by TEM analysis for the GaO_x nanosheets grown by the PA-LMP with 30°C and 0.3 kgf/cm², which was presented as **Figure R7 in our response in Reviewer 1.**

Comment 6. *Actually, when we used the squeezing method to fabricate 2D Ga₂O₃, it also exist an assisted pressure approaching 0.3 kgf/cm². However, the fabricated 2D Ga₂O₃ film is insulated. So, what's the key point and technology of your proposed PA-LMP method to induce the crystallization of such thin 2D-Ga₂O₃ layer? How about the repeatability or success rate of the experiment in Ga₂O₃ layer crystallization and device? It is suggested to provide a video of preparation process of PA-LMP.*

(Response) Thank you for your feedback and questions. Both the process temperature and time duration are also key factors in the growth of semiconducting Ga₂O₃ in the PA-LMP method. If the process temperature is too high or the processing time is too long, the Ga₂O₃ exhibits insulative behavior, and the TFT device does not work. This is primarily attributed to low electron density in Ga₂O₃. It is, in general, known that electron density is very critical for TFT device operation. Therefore, controlling process temperature and time is necessary. Please also refer to the discussion of the effect of process time in **Review1 comment 13** and the effect of the process temperature in our manuscript **Figure 3**.

For the repeatability of Ga₂O₃ layer crystallization and device fabrication; The yields of TFT operable devices fabricated at a higher pressure of 1.3 kgf/cm² is not as high as those fabricated with low pressure at 0.3 kgf/cm². To enhance repeatability and uniformity, we are currently actively optimizing the printing process. The optimization of the printing process, including the uniformity of pressed pressure and control of the experimental environment with controllable oxygen concentration and humidity, etc., would be important for further improvement in yield and uniformity in the future. Please refer to our response in **Comment 10 in Reviewer 1** for the batch-to-batch

variation of Ga₂O₃ nanosheet TFTs (**Figure R15**). In addition, we have addressed the uniformity of Ga₂O₃ TFT performance in our response to the **Comment 10 in Reviewer 1**). We have included data and discussions about uniformity and batch-to-batch variations of device performance in the supporting information in the revised manuscript.

Comment 7. *Have the authors tried higher assisted pressure than 1.3 kgf/cm² and higher post-annealing temperature than 400 °C in the preparation process? How it will affect the crystal quality and electrical performance of 2D Ga₂O₃ film?*

(Response) Figure R26 (Figure S32 in the supporting information in the revised manuscript) summarizes the transfer characteristics of the Ga₂O₃ TFTs fabricated under different process pressures and temperature conditions during the pressure-assisted printing route.

Summary: (1) Ga₂O₃ films grown at high pressures of 2.1 kgf/cm² (larger than 1.3 kgf/cm²) with the T_p of 150 and 200 °C exhibited conductive behavior with weak *p*-type characteristics.

(2) The TFT performance degraded with a positive threshold voltage shift during the high-temperature process at 200 °C under the P_p of 0.3 and 1.3 kgf/cm², respectively. We suspect this phenomenon primarily results from the decrease of carrier concentration, originating from the oxygen vacancies, which can be decreased through compensation by oxygen-containing high-temperature processes in ambient air.

In response to the reviewer's suggestion, we also explored the T_p of 300, 400, and 450 °C (**Figure R27**). However, all the devices exhibited insulating behavior.

We have added more detailed discussions in the revised manuscript. The revised section in the manuscript: "The TFT performance degraded with a positive threshold voltage shift during the high-temperature process at 200 °C under pressures of 0.3 and 1.3 kgf/cm², respectively. The reason is suspected to be a decrease in carrier concentration originating from oxygen vacancies."

Figure R26. Variation of transfer characteristics for the Ga_2O_3 TFTs fabricated by the PA-LMP with different temperatures and pressure conditions during the printing process.

Figure R27. Transfer characteristics for the Ga₂O₃ nanosheet TFTs with different printing process temperatures under the uniaxial process pressure of 0.3 kgf/cm². (a) 300 °C, (b) 400 °C, and (c) 450 °C. Transfer characteristics for the Ga₂O₃ nanosheet TFTs with different printing process temperatures under the uniaxial process pressure of 1.3 kgf/cm². (a) 300 °C, (b) 400 °C, and (c) 450 °C.

Comment 8. *How about the continuity of the film? Does it exist cracks and holes? Please give low-magnification optical image and more description in the main text.*

(Response) To address the reviewer’s question, we have provided the optical microscope images of large-scale ($1 \times 1 \text{ cm}^2$) Ga₂O₃ nanosheets fabricated by pressure-assisted liquid metal printing at the T_p of 150 °C under different pressures in **Figure R20**. We confirmed that the entire laterally large Ga₂O₃ layer is continuous and without significant holes and cracks. **Please refer to our response to thee Comment 12 in Reviewer 1.**

Following the reviewer's suggestion, we have included a low-magnification optical image and provided additional descriptions in the main text in the revised manuscript. Additionally, we have included **Figure R20** in the supporting information (**Figure S4**).

The revised section in the manuscript: "The optical microscope images and photographs of the Ga₂O₃ grown by previously reported liquid metal printing (LMP) and the proposed PA-LMP method show that these samples exhibit large-scale Ga₂O₃ nanosheets with dimensions of around 1×1 cm². (**Figure 1 (a), (b)**) (The schematic of the conventional LMP is shown in **Figure S3**, , Supporting Information.) The entire laterally large Ga₂O₃ layer is continuous and without significant holes and cracks under optimized synthesis operations. (**Figure S4** (Supporting Information) shows additional optical microscope images of the Ga₂O₃ nanosheet fabricated by the different printing process parameters.)"

"The optical microscope images of these samples with different process conditions, which we investigated for their electrical properties, are also shown in **Figure S4**. (Supporting Information)"

Comment 9. *Are there any tiny liquid Ga droplets (normally < 5nm) on the surface of 2D Ga₂O₃ film? High-resolution SEM observation is required to check it.*

(Response) Following the reviewer's suggestion, we conducted an SEM observation. No liquid Ga droplets on the surface of the Ga₂O₃ film were confirmed, as shown in **Figure R28**. The presence of relatively large particles is not liquid Ga droplets but dust generated in SEM sample preparation in a non-controlled ambient atmosphere.

Moreover, we observed no effect of post-air annealing on TFT device performances. (**Figure S6(a)** in supporting information), implying that Ga droplets unlikely to degrade TFT devices. Therefore, we would also like to note that the Ga₂O₃ TFTs are not largely impacted by tiny liquid Ga droplets if they are present.

Figure R28. SEM image of the surface of the Ga₂O₃ channel.

Comment10. Why the authors choose ITO as the materials of source and drain electrodes? Have you tried Ti/Au metal electrodes? Does it influence the performance of TFTs?

(Response) The formation of ohmic contacts is an essential factor for TFT performances. We examined ITO (40 nm) and Ti/Au (20/20 nm) as the source/drain electrode for β -Ga₂O₃ TFTs. The ITO was deposited by pulsed laser deposition (PLD), while the Ti/Au was deposited by e-beam evaporation. **Figure R29** summarized their TFT characteristics. The Ti/Au contact exhibited non-linear behavior in the output characteristics and degraded the on-current property. In contrast, the good ohmic relation was observed in ITO contact. The literature reports that ITO makes good contact with Ga₂O₃ with improved contact resistance.²⁰⁻²² Therefore, we utilized ITO as the source/drain electrode for our β -Ga₂O₃ TFT device.

The revised section in the manuscript: “The ITO was used as the source/drain electrode for the Ga₂O₃ TFTs to form ohmic contacts.”

Figure R29. (a) Transfer characteristics for linear ($V_{DS} = 1$ V) and saturation region ($V_{DS} = 20$ V) of the β -Ga₂O₃ nanosheet TFTs using ITO contact. (b) corresponding

output characteristics for the β -Ga₂O₃ TFT. (c) Magnification of the linear regime for the output characteristics. (d) Transfer characteristics for linear ($V_{DS}=1V$) and saturation region ($V_{DS}=20V$) of the β -Ga₂O₃ nanosheet TFTs using Au contact. (e) corresponding output characteristics for the Ga₂O₃ TFT. (f) Magnification of the linear regime for the output characteristics.

Comment 11. *The optical photograph and SEM images of TFTs should be added in the supporting information.*

(Response) Following to the reviewer's suggestion, we observed the β -Ga₂O₃ channel by optical microscopy and SEM. **(Figure R30)** We added **Figure R30** to the supporting information and the following sentences in the manuscript. "The optical microscopy image and scanning electron microscopy of the Ga₂O₃ TFT are shown in **Figure S6**. (Supporting Information)"

Figure R30. (a) Optical microscopy image and (b) Scanning electron microscopy image of the β -Ga₂O₃ TFT.

Comment 12. *How about the stability of 2D Ga₂O₃ film and related TFTs performance in the ambient air atmosphere after one week or one month?*

(Response) Following the reviewer's suggestion, we investigated the environmental stability of the β -Ga₂O₃ TFTs. In **Figure R31**, the transfer characteristics of the β -Ga₂O₃ TFTs are presented for the as-fabricated devices and those stored in ambient air atmosphere for 2 weeks and 1 month after the device fabrication. No significant

degradations were observed in all devices, indicating that the presented devices had good environmental stability.

We have added these data to the Supporting Information and the following sentences in the manuscript. “Furthermore, we observed no significant degradation in the device characteristics of the $\beta\text{-Ga}_2\text{O}_3$ TFTs even after 1 month of storage under a non-controlled ambient atmosphere, confirming the environmental stability of the devices in the air. (Figure S10 (Supporting Information) for the environmental stability test for $\beta\text{-Ga}_2\text{O}_3$ TFTs)”

Figure R31. Variation of transfer characteristics for the as-fabricated $\beta\text{-Ga}_2\text{O}_3$ -TFT devices and those after storage in ambient air atmosphere for 2 weeks and 1 month after fabrication: (a) Device 1, (b) Device 2, (c) Device 3, respectively.

Comment 13. *How about the photoresponse of 2D $\beta\text{-Ga}_2\text{O}_3$ film? It is also an important property of 2D Ga_2O_3 film. Does it only respond to deep ultraviolet light?*

(Response) Following the reviewer’s suggestion, we also have examined the photoresponse of the $\beta\text{-Ga}_2\text{O}_3$ TFTs. (T_p of $150\text{ }^\circ\text{C}$ and P_p of 1.3 kgf/cm^2) The measurements were performed under green, blue, and UV light illuminations using the respective LEDs: green LED ($\lambda=520\text{ nm}$, Power density= 0.1 mW/cm^2), blue LED ($\lambda=460\text{ nm}$, Power density= 0.15 mW/cm^2), and UV LED ($\lambda=395\text{ nm}$, Power density= 0.1 mW/cm^2). (Figure R32) No photoresponse under green, blue, or UV light illuminations were observed. The deep ultraviolet light response of the presented $\beta\text{-Ga}_2\text{O}_3$ TFTs is our future important project.

The revised section in the manuscript, “The photoresponse property of β -Ga₂O₃ TFTs was also evaluated. No photoresponse was observed under green, blue, or UV light illuminations, which can be attributed to the wide-bandgap nature of β -Ga₂O₃. (Figure S11, Supporting Information)”

Figure R31. Transfer characteristics for the β -Ga₂O₃ TFTs (T_p of 150 °C and P_p of 1.3 kgf/cm²) without and with green light illumination ($\lambda=520$ nm, Power density=0.1 mW/cm²): (a) Device 1, (b) Device 2, and (c) Device 3. Transfer characteristics for the β -Ga₂O₃ TFTs without and with blue LED ($\lambda=460$ nm, Power density=0.15 mW/cm²): (d) Device 1, (e) Device 2, and (f) Device 3. Transfer characteristics for the β -Ga₂O₃ TFTs without and with UV LED ($\lambda=395$ nm, Power density=0.1 mW/cm²): (g) Device 1, (h) Device 2, and (i) Device 3.

Reference

1. Hamlin, A.B., Ye, Y., Huddy, J.E., Rahman, M.S. & Scheideler, W.J. 2D transistors rapidly printed from the crystalline oxide skin of molten indium. *npj 2D Materials and Applications* **6**, 16 (2022).
2. Jannat, A. et al. Printable Single-Unit-Cell-Thick Transparent Zinc-Doped Indium Oxides with Efficient Electron Transport Properties. *ACS nano* **15**, 4045-4053 (2021).
3. Tang, Y., Huang, C.-H. & Nomura, K. Vacuum-Free Liquid-Metal-Printed 2D Indium–Tin Oxide Thin-Film Transistor for Oxide Inverters. *ACS nano* **16**, 3280-3289 (2022).
4. Li, Q. et al. Gas-mediated liquid metal printing toward large-scale 2D semiconductors and ultraviolet photodetector. *npj 2D Materials and Applications* **5**, 36 (2021).
5. Lee, S. & Nathan, A. Subthreshold Schottky-barrier thin-film transistors with ultralow power and high intrinsic gain. *Science* **354**, 302-304 (2016).
6. Jiang, C. et al. Printed subthreshold organic transistors operating at high gain and ultralow power. *Science* **363**, 719-723 (2019).
7. Wang, G. et al. New Opportunities for High-Performance Source-Gated Transistors Using Unconventional Materials. *Advanced Science* **8**, 2101473 (2021).
8. Hsieh, H.-H., Kamiya, T., Nomura, K., Hosono, H. & Wu, C.-C. Modeling of amorphous InGaZnO₄ thin film transistors and their subgap density of states. *Appl. Phys. Lett.* **92** (2008).
9. Korhonen, E. et al. Electrical compensation by Ga vacancies in Ga₂O₃ thin films. *Appl. Phys. Lett.* **106** (2015).
10. Varley, J.B., Peelaers, H., Janotti, A. & Van de Walle, C.G. Hydrogenated cation vacancies in semiconducting oxides. *J. Phys.: Condens. Matter* **23**, 334212 (2011).
11. Kim, J. et al. Conversion of an ultra-wide bandgap amorphous oxide insulator to a semiconductor. *NPG Asia materials* **9**, e359-e359 (2017).
12. Syed, N. et al. Wafer-Sized Ultrathin Gallium and Indium Nitride Nanosheets through the Ammonolysis of Liquid Metal Derived Oxides. *J. Am. Chem. Soc.* **141**, 104-108 (2019).
13. Li, J. et al. Template Approach to Large-Area Non-layered Ga-Group Two-Dimensional Crystals from Printed Skin of Liquid Gallium. *Chem. Mater.* **33**, 4568-4577 (2021).
14. Datta, R.S. et al. Flexible two-dimensional indium tin oxide fabricated using a liquid metal printing technique. *Nat. Electron.* **3**, 51-58 (2020).

15. Das, M., Chakraborty, T., Lin, C.Y., Lin, R.-M. & Kao, C.H. Screen-printed Ga₂O₃ thin film derived from liquid metal employed in highly sensitive pH and non-enzymatic glucose recognition. *Mater. Chem. Phys.* **278**, 125652 (2022).
16. Ghasemian, M.B. et al. Ultra-thin lead oxide piezoelectric layers for reduced environmental contamination using a liquid metal-based process. *Journal of Materials Chemistry A* **8**, 19434-19443 (2020).
17. Messalea, K.A. et al. Bi₂O₃ monolayers from elemental liquid bismuth. *Nanoscale* **10**, 15615-15623 (2018).
18. Zavabeti, A. et al. High-mobility p-type semiconducting two-dimensional β -TeO₂. *Nat. Electron.* **4**, 277-283 (2021).
19. Zhang, Y., Huang, C.-H. & Nomura, K. High-mobility wide bandgap amorphous gallium oxide thin-film transistors for NMOS inverters. *Applied Physics Reviews* **11** (2024).
20. Oshima, T. et al. Formation of indium–tin oxide ohmic contacts for β -Ga₂O₃. *Jpn. J. Appl. Phys.* **55**, 1202B1207 (2016).
21. Carey, P.H., IV et al. Improvement of Ohmic contacts on Ga₂O₃ through use of ITO-interlayers. *Journal of Vacuum Science & Technology B* **35** (2017).
22. Pearton, S.J. et al. A review of Ga₂O₃ materials, processing, and devices. *Applied Physics Reviews* **5** (2018).

Response to the reviewer's comments and the list of corrections

Manuscript ID: NCOMMS-23-45192

Title: Low-temperature crystallization enabled by pressure-assisted liquid-metal printing for β -Ga₂O₃ thin-film transistors

Authors: Chi-Hsin Huang, Ruei-Hong Cyu, Yu-Lun Chueh and Kenji Nomura*

Reviewer #1 (Remarks to the Author):

The authors have addressed the majority of the comments from the first round of review.

I have a few other points that need resolution:

1) There is no reasonable reason to use kgf as the units for the pressure. Just use Pa / kPa, etc. Kgf will confuse readers unnecessarily since it is a non-standard unit

(Response) Thank you for review's suggestion. We have modified the units of pressure to kPa in the revised manuscript.

2) The low power consumption is really not a unique advantage of this material compared to any other standard TFT technology. The authors severely overstate the capabilities in terms of low power consumption. Comparing it against silicon CMOS is not a good way to go – standard MOSFET devices could easily be made with low power consumption. The existence of the entire field of low power VLSI systems alone should prove that the authors argument is a bit of a reach here. I would like to see removal or at the very least the tempering of the claims to low power logic operation. The devices are good but not necessarily unique compared to the state of the art in metal oxide TFTs and other low power devices. The discussion on pg. 25 should be updated to reflect this. All that being said, the authors do certainly show low off-state currents. This is a good thing. The authors should have written the discussion of this fact to cite the literature on IGZO displays, which highlights their off currents as an advantage. I recommend doing this rather than trying to rationalize whether 84 nW is a significantly lower power consumption.

(Response) Thank you for the review’s constructive comments and suggestions. Following review’s request, we have removed the sentence, “Since power consumption can be further reduced through scaling, the presented results demonstrate high potential for ultra-low power consumption TFT technology. This is attributed to the low off-current characteristics originating from the ultra-wide bandgap nature.” for the NMOS inverter part and “These results reveal the potential for achieving low-power operation for oxide CMOS circuits using scaled devices. The low-power oxide-CMOS inverter using p -SnO/ n -Ga₂O₃ TFTs is attributed to the ultralow off-current nature for both the n - and p -channel transistors.” for the CMOS inverter part.

Moreover, following the review’s suggestion, we have added the following sentence and cited the reference to discuss the off-current comparison with current IGZO TFT technology in the display applications in the revised manuscript. “In addition, the off-current density of the presented TFT devices is estimated to be approximately $\sim 3 \times 10^{-15}$ A/ μm , limited by the measurement instrument. The leakage current of the Ga₂O₃ TFTs is potentially even lower than that of commercial a-IGZO TFT devices ($\sim 10^{-18}$ A/ μm)^{1,2} in display applications. This is attributed to the wider bandgap nature of the Ga₂O₃ channel compared to a-IGZO.”

3) *On pg. 13 the authors mention that the density of their amorphous GaO_x was observed to be lower?? Did they measure it with x-ray methods or are they citing the literature shown there? It is unclear and misleading.*

(Response) Thank you for pointing out the confusing part. We referenced literature to obtain the film density of amorphous GaO_x. In response to the review’s suggestion, we have revised the following sentence and added references for the film density of amorphous GaO_x and β -Ga₂O₃, respectively. “Based on previously reported experimental results, the film density of amorphous GaO_x was determined to be approximately 5.2–5.4 g·cm⁻³,^{3,4} which is lower than that of β -Ga₂O₃ (density of ~ 5.95 g·cm⁻³).⁵⁻⁷”

4) *Do the authors have any evidence that the gap at these ranges of pressure is at a sufficiently small length scale to result in the confined crystallization hypothesized? I would want to see some additional evidence of this claim before they cite the SI discussion on 2D confined crystallization.*

(Response) Thank you for the constructive suggestions. We have reviewed and summarized previous studies on crystallization phenomena under nanoscale confinement.

Kapadia *et al.*⁸ developed the thin-film vapor-liquid-solid (TF-VLS) growth technique to produce polycrystalline InP thin films with grain sizes ranging from 10 to 100 μm and a thickness of approximately 3 μm on non-epitaxial surfaces without the need for nucleation seeding. In this process, indium was deposited onto a polished Mo foil and capped with a SiO_x layer to create a confined geometry. The structure was then annealed in a phosphorus-rich environment at a temperature above the melting point of indium. Phosphorus diffused through the SiO_x layer and mixed with the liquid indium, resulting in InP nucleation and subsequent growth from the liquid solution.

Building on this concept, Chen *et al.*⁹ developed the templated liquid-phase crystal growth (TLPCG) technique. In this method, MoO_x -In- SiO_x stacks are lithographically patterned on glass or Si/ SiO_2 substrates and encapsulated by SiO_x within a confined geometry. Following this, annealing at the temperature above the melting point of indium in a phosphorus environment within the templated confinement results in the formation of InP thin films. Under specific growth conditions, single-crystal InP films with cylindrical shapes and diameters up to 7 μm were achieved. Both the TF-VLS and TLPCG techniques, which are governed by heterogeneous nucleation—where nuclei form at an interface within the confined film—facilitate the growth of large-grained thin films by modulating nucleation energy barriers.

Mastandrea *et al.*¹⁰ applied classical nucleation theory to investigate nucleation during the solidification and melting phase transitions within nanoscale confinement between two planar surfaces. They calculated the critical nucleus shapes and corresponding nucleation energy barriers for thin films confined in the nanogap, considering film thickness and relevant bulk and interface energies. The study used this theory to predict the temperatures at which nucleation of the solid and liquid phases occurs for Ge between two glass substrates. When germanium (Ge) is confined between these glass substrates, its nucleation temperature decreases significantly for both solid and liquid phases. According to their findings, with a confinement gap of 5.0 nm, the nucleation temperature of Ge can be reduced by as much as 350°C.

Similar findings have been reported in previous studies. In 1986, Takagi derived a theoretical relationship for phase transition temperatures in confined thin films, showing how they depend on film thickness and bulk/interface energies.¹¹ This theory

was applied to calculate the melting temperature depression for Pb and Au. Zhou *et al.*¹² later used molecular dynamics (MD) simulations to investigate the solidification of aluminum (Al) melt in confined nanoslits (< 4 nm) formed by two substrates. Compared to a single substrate, the nanoslits promoted crystallization, with the size of the slit having a significant impact on the structure of the solidified material.

In addition, metastable phases have been observed to crystallize in inorganic compounds under nanoscale confinement and stress. For example, pressure injection was used to fill the pores of an anodic aluminum oxide (AAO) membrane with liquid bismuth (Bi), which then crystallized during slow cooling.¹³ Since Bi expands during its transformation from liquid to solid, it is suggested that the membrane pores impose external stress, stabilizing the high-pressure crystal structure.

Following the review's suggestion, we have added the following sentence to the revised manuscript. "Previous studies have explored crystallization and phase transitions under nanoscale confinement.¹⁰⁻¹² For example, Mastandrea *et al.*¹⁰ applied classical nucleation theory to investigate nucleation during the solidification and melting phase transitions of germanium (Ge) within nanoscale confinement between two planar surfaces. They found that nanoscale confinement significantly lowers nucleation temperatures of Ge for solid and liquid phases, with a 5.0 nm gap reducing nucleation temperature by up to 350°C."

Others: There are still a number of minor grammatical issues to resolve. Here are a couple of examples. These need to be fixed.

(Response) We have addressed the grammatical issues in the revised manuscript.

(Response) "The static currents are lower than 1 nA, leading to the low static power dissipation is 20 nW per logic gate at VDD of 50V"

We have corrected the grammar of this sentence. "The static currents are lower than 1 nA, leading to the low static power dissipation of 20 nW per logic gate at a VDD of 50V."

(Response) "The low-power oxide-CMOS inverter using p-SnO/n-Ga2O3 TFTs is attributed to the ultralow off-current nature for both the n- and p-channel transistors."

We have removed this sentence.

Reviewer #2 (Remarks to the Author):

While the authors have conducted a lot of extra work, they failed to genuinely address and engage with the core criticism that was raised. A pressure of 1.3 kgf/cm² (or ~1.3 bar) is inconsequential. There is just not enough energy applied to impact crystallization. Let's be clear, I believe that the authors made high performing devices from LM printed Ga₂O₃. I might also be convinced that the better performance is linked to changes in crystallinity. I just don't believe that the applied pressure has anything to do with the change in performance / crystallinity. Something else must be going on that impacts crystal growth / electronic behaviour. If the paper is published as is there is a high chance that the core mechanism that is put forward is incorrect. As such, the authors should really put forward a strong case regarding how such small pressure can have an effect. This needs to go beyond showing that there are changes in crystallinity. It needs to link these changes to the tiny increase in pressure.

Aside from this, the authors also did not show that the material is actually crystalline - Yes, they show crystalline domains but there is no evidence that the material is crystalline beyond the scale of the TEM images - i.e. ~10 to 20 nm. The material may be amorphous with occasional crystalline inclusions/domains.

Raman mapping would go a long way. Alternatively a systematic study on a larger sample of 2D Ga₂O₃ in the TEM with many sections being measured at different locations would help. This would have to be done for both, the crystalline sample prepared under pressure and the amorphous sample prepared without pressure. I do acknowledge that the cross sectional samples are moving towards this, but as it stands these would need to be supplemented.

The GIXRD measurements are not that useful at this point. As the authors point out, getting x-ray diffraction data on thin films is extremely difficult. The indicated peaks are as far as I can tell in the noise level and cannot be identified with confidence. It is also not possible to tell if these peaks (if they are there) arise from the 2D film or from thick inclusions.

I do want to highlight that the work on the electronic device aspects has been significantly strengthened and expanded. These aspects are definitely worthy of publication. The synthesis aspects and the proposed mechanism are the issue.

(Response) Thank you for the constructive comments and suggestions. In response to the review's request, we performed cross-sectional TEM analysis for both crystallized Ga₂O₃ and amorphous GaO_x nanosheets at various locations. **Figure R1(a)** shows a

cross-sectional low-magnification TEM image of a sample fabricated using a focused ion beam (FIB). The sample is the Ga₂O₃ nanosheet synthesized by the PA-LMP route at temperatures of 150 °C and a pressure of 1.3 kgf/cm². (T_p of 150 °C and P_p of 129 kPa) **Figure R1(b)** presents high-magnification cross-sectional TEM images from different regions of the Ga₂O₃ nanosheet from different regions, indicated by rectangles (1), (2), and (3) in **Figure R1(a)**. **Figure R1(c)** shows the SAED patterns corresponding to regions (1), (2), and (3) in **Figure R1(a)**, indexed to the planes of β-Ga₂O₃. These results confirm that the Ga₂O₃ nanosheet, prepared by pressure-assisted liquid-metal printing, exhibits a crystalline β-Ga₂O₃ phase consistently across various locations over a large area (on the order of a few μm).

For comparison, **Figure R2(a)** shows a low-magnification cross-sectional TEM image of a Ga₂O₃ nanosheet synthesized *via* the LMP route at 150°C without applied pressure (T_p = 150°C, P_p = 0 kPa), with the sample fabricated using FIB tool. **Figure R2(b)** presents high-magnification cross-sectional TEM images of the Ga₂O₃ nanosheet from different regions, indicated by rectangles (1), (2), and (3) in **Figure R2(a)**, revealing no lattice-ordered structure in the LMP-grown nanosheet. **Figure R2(c)** shows the corresponding SAED patterns from rectangles (1), (2), (3), and (4) in **Figure R2(a)**, confirming the presence of a halo pattern. These results indicate that the GaO_x nanosheet, prepared by liquid-metal printing without applied pressure, remains in an amorphous phase consistently across various locations over a large area (on the order of a few μm).

Therefore, we have added the data to the supporting information (**Figure S22 and Figure S23**).

Figure R1. (a) Low-magnification cross-sectional TEM image of a sample fabricated using FIB for the Ga_2O_3 nanosheet synthesized by the PA-LMP route with a T_p of 150 °C and a P_p of 129 kPa. (b) Corresponding cross-sectional HRTEM image. (c) Corresponding SAED patterns from regions (1), (2), and (3), respectively.

Figure R2. (a) Low-magnification cross-sectional TEM image of a sample fabricated using FIB for the GaO_x nanosheet was synthesized using the conventional liquid-metal printing route. (without applied printing process pressure) (b) Corresponding cross-sectional HRTEM image. (c) Corresponding SAED patterns from regions (1), (2), and (3), respectively.

Reviewer #3 (Remarks to the Author):

The revisions can address our concerns in full and the revised manuscript is suitable for publication.

Reference

1. Hosono, H. & Kumomi, H. Amorphous Oxide Semiconductors: IGZO and Related Materials for Display and Memory. (John Wiley & Sons, 2022).
2. Kim, T. et al. Progress, Challenges, and Opportunities in Oxide Semiconductor Devices: A Key Building Block for Applications Ranging from Display Backplanes to 3D Integrated Semiconductor Chips. *Adv. Mater.* **35**, 2204663 (2023).
3. Kim, J. et al. Conversion of an ultra-wide bandgap amorphous oxide insulator to a semiconductor. *NPG Asia materials* **9**, e359-e359 (2017).
4. Zhang, Y., Huang, C.-H. & Nomura, K. High-mobility wide bandgap amorphous gallium oxide thin-film transistors for NMOS inverters. *Applied Physics Reviews* **11** (2024).
5. Geller, S. Crystal structure of β -Ga₂O₃. *The Journal of Chemical Physics* **33**, 676-684 (1960).
6. Åhman, J., Svensson, G. & Albertsson, J. A reinvestigation of β -gallium oxide. *Acta Crystallogr. Sect. C: Cryst. Struct. Commun.* **52**, 1336-1338 (1996).
7. Poncé, S. & Giustino, F. Structural, electronic, elastic, power, and transport properties of β -Ga₂O₃ from first principles. *Physical Review Research* **2**, 033102 (2020).
8. Kapadia, R. et al. A direct thin-film path towards low-cost large-area III-V photovoltaics. *Sci. Rep.* **3**, 2275 (2013).
9. Chen, K. et al. Direct growth of single-crystalline III-V semiconductors on amorphous substrates. *Nat. Commun.* **7**, 10502 (2016).
10. Mastandrea, J.P., Ager, J.W., III & Chrzan, D.C. Nucleation of melting and solidification in confined high aspect ratio thin films. *J. Appl. Phys.* **122** (2017).
11. Takagi, M. The Thickness Dependence of the Phase Transition Temperature in Thin Solid Films. *J. Phys. Soc. Jpn.* **55**, 3484-3487 (1986).
12. Zhou, X. et al. Heterogeneous nucleation of Al melt in symmetrical or asymmetrical confined nanoslits. *Nanoscale* **8**, 12339-12346 (2016).
13. Zhang, Z., Gekhtman, D., Dresselhaus, M.S. & Ying, J.Y. Processing and Characterization of Single-Crystalline Ultrafine Bismuth Nanowires. *Chem. Mater.* **11**, 1659-1665 (1999).

Response to the reviewer's comments and the list of corrections

Manuscript ID: NCOMMS-23-45192B

Title: Low-temperature pressure-assisted liquid-metal printing for β -Ga₂O₃ thin-film transistors

Authors: Chi-Hsin Huang, Ruei-Hong Cyu, Yu-Lun Chueh and Kenji Nomura*

Reviewer #2 (Remarks to the Author):

Once again I feel that the authors have not genuinely attempted to show that such a low pressure can indeed cause crystallisation. To be clear - I do not doubt that crystallisation occurs. 129 kpa is just not enough. For comparison, this pressure is actually on the lower end of what is found in soda cans. It is just not convincing that this low pressure can cause changes in crystallisation habit. The chemistry does not make sense. When we conduct similar experiments in our lab and apply pressure to the 2D GaOx film we do not see the reported crystallisation, however we did not spend as much time on this as the authors have.

Having said the above. I feel that after 2 years of revisions the editor needs to make a final call. The authors are clearly convinced that they got it right. I don't think that I can be convinced. Maybe it is time to just publish the work and if people can reproduce it then I am happy to be proven wrong. Maybe in a future paper a different mechanism is proposed that is more viable. Or, the paper is found to not be reproducible and the field moves on.

Edit guidance:

In view of the remaining concerns raised by referee #2, please moderate your claims about the influence of pressure on the crystallization of your β -Ga₂O₃ thin films. Please revise the Title and Abstract as suggested below. Moreover, please use phrases such as "suggests, appears to.." etc in every discussion about the role of pressure in the investigated synthesis/crystallization process. Please also clearly state in the Main Text that further work will be necessary to confirm the influence of pressure on β -Ga₂O₃ crystallization.

(Response) We are grateful for the opportunity to revise our manuscript and sincerely appreciate the reviewers' valuable comments, as well as the editor's insightful guidelines and suggestions. In response to the reviewers' request:

1. Following the editor's suggestion, we have revised the Title and Abstract accordingly. New title is "Low-temperature pressure-assisted liquid-metal printing for β -Ga₂O₃ thin-film transistor"

2. We have carefully revised our discussions throughout the manuscript to moderate our claims about the role of pressure in the crystallization of β -Ga₂O₃. Accordingly, we have rewritten the following sentences in the main text of the revised manuscript:

"Furthermore, our findings suggest that the effects of nanoscale confinement crystal growth and external pressure may influence the facilitation of low-temperature crystallization for the β -Ga₂O₃ phase."

"As a result, our findings suggest that pressure may also facilitate crystallization by potentially reducing the nucleation activation energy."

3. We have explicitly stated in the main text that further investigation is required to confirm the influence of pressure on β -Ga₂O₃ crystallization. We have added the following sentence in the revised manuscript:

"Further research is required to validate the impact of applied pressure on the β -Ga₂O₃ crystallization process."